# Enhanced Arctic sea ice melting controlled by larger heat discharge of mid-Holocene rivers

Jiang Dong [1,11], Xuefa Shi [1,2,11] ✉, Xun Gong [3,4,5,6,11] ✉, Anatolii S. Astakhov[7], Limin Hu[2,8], Xiting Liu [2,8], Gang Yang[1], Yixuan Wang[9], Yuri Vasilenko [7], Shuqing Qiao[1,2], Alexander Bosin [7] & Gerrit Lohmann [6,10]

Arctic sea ice retreat is linked to extrapolar thermal energy import, while the potential impact of pan-Arctic river heat discharge on sea-ice loss has been unresolved. We reconstructed the Holocene history of Arctic sea ice and Russian pan-Arctic river heat discharge, combining ice-rafted debris records and sedimentation rates from the East Siberian Arctic Shelf with a compilation of published paleoclimate and observational data. In the mid-Holocene, the early summer (June–July) solar insolation was higher than that during the late Holocene, which led to a larger heat discharge of the Russian pan-Arctic rivers and contributed to more Arctic sea ice retreat. This intensified decline of early-summer sea ice accelerated the melting of sea ice throughout the summertime by lowering regional albedos. Our findings highlight the important impact of the larger heat discharge of pan-Arctic rivers, which can reinforce Arctic sea-ice loss in the summer in the context of global warming.

Arctic sea ice has dramatically declined under global warming, with feedback accelerating ongoing warming conditions[1–4]. However, future climate trends in association with sea ice melting remain elusive due to an insufficient understanding of the mechanism of retreat or even the absence of summer Arctic sea ice in the future. By referring to the Arctic climate of warmer intervals in the past, paleoclimate studies can provide clues for the future; e.g., the mid-Holocene (MH) was the most recent warmer climate interval relative to modern conditions.

Previous studies have attributed the changes in the Arctic sea ice during the Holocene to solar insolation[5,6], poleward moisture in association with latent heat transport[1,7], and the Atlantic[8,9] and Pacific

inflows[10] of relatively warm water (Fig. 1a). Additionally, based on limited instrumental records[11,12] and one numerical simulation[13], studies have also shown that pan-Arctic rivers have delivered heat from continents into the Arctic Ocean via the discharge of warming waters in recent decades (Fig. 1b, c). In early summer (June–July), with the seasonal peak in the solar energy input[14–16], the thermal flux of modern pan-Arctic rivers can also be sourced from the high and middle latitudes, depending on river basin-wide heat regimes and areas[15], approximately equivalent to 10% of the sum of the heat input via Atlantic and Pacific ocean water into the Arctic Ocean[17]. Although the pan-Arctic river heat flux is relatively small, the discharge is

[1]Key Laboratory of Marine Geology and Metallogeny, First Institute of Oceanography, Ministry of Natural Resources, Qingdao, China. [2]Laboratory for Marine Geology, Pilot Qingdao National Laboratory for Marine Science and Technology, Qingdao, China. [3]Institute for Advanced Marine Research, China University of Geosciences, Guangzhou, China. [4]State Key Laboratory of Biogeology and Environmental Geology, Hubei Key Laboratory of Marine Geological Resources, China University of Geosciences, Wuhan, China. [5]Shandong Provincial Key Laboratory of Computer Networks, Qilu University of Technology (Shandong Academy of Sciences), Jinan, China. [6]Alfred-Wegener-Institut Helmholtz-Zentrum für Polar- und Meeresforschung, Bremerhaven, Germany. [7]V.I.Il'ichev Pacific Oceanological Institute, Far Eastern Branch of Russian Academy of Sciences, Vladivostok, Russia. [8]Key Laboratory of Submarine Geoscience and Prospecting Techniques, College of Marine Geosciences, Ocean University of China, Qingdao, China. [9]Key Laboratory of Comprehensive and Highly Efficient Utilization of Salt Lake Resources, Qinghai Institute of Salt Lakes, Chinese Academy of Sciences, Xining, China. [10]Department of Environmental Physics, University of Bremen, Bremen, Germany. [11]These authors contributed equally: Jiang Dong, Xuefa Shi, Xun Gong. ✉e-mail: xfshi@fio.org.cn; gongxun@cug.edu.cn

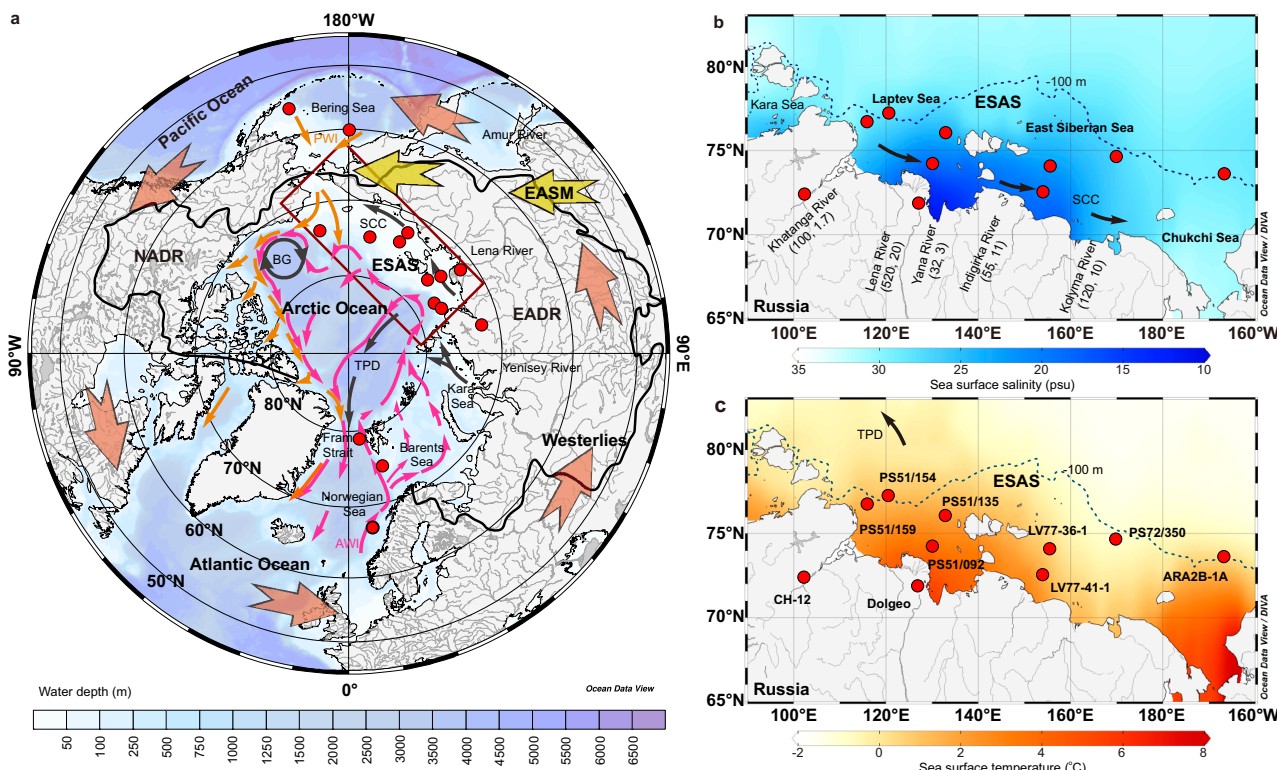

**Fig. 1 | Map of the modern pan-Arctic region with insets of the East Siberian Arctic Shelf (ESAS) area. a** Pan-Arctic river system, Arctic Ocean circulation (arrow lines)[17,24], Northern Hemisphere westerlies (orange arrows), and East Asian summer monsoon (EASM, yellow arrows). Sea surface (**b**) salinity and **c** temperature (color scales) during July–September in the ESAS region, with mean annual Russian pan-Arctic major river runoff and sediment load (km³/yr and Mt/yr, respectively)[23]. The red dots show the locations of sediment cores. EADR, NADR, PWI, AWI, TPD, BG, and SCC are the Eurasian pan-Arctic discharge realm, North American pan-Arctic discharge realm, Pacific water inflow at a water depth of 40–220 m, Atlantic water inflow at a water depth of 200–800 m, surface Transpolar Drift, surface Beaufort Gyre, and surface Siberian Coastal Current, respectively.

concentrated, and warm freshwater can directly contact Arctic sea ice because of buoyancy. Compared to the present climate, the Arctic climate in MH summer became warmer and had less sea ice[10,18], along with stronger solar insolation in the Northern Hemisphere[19–22]. As indicated by the modern global warming process[13,15], a stronger thermal flux associated with discharge water by pan-Arctic rivers would have contributed to the loss of sea ice in the MH summer due to the higher-than-present solar insolation. However, such a potential physical mechanism for the MH climate has been ignored until this study.

Under the modern climate, the East Siberian Arctic Shelf (ESAS, where the water depth is shallower than 100 m in the Laptev Sea, East Siberian Sea, and Chukchi Sea) is a prominent area of increasing sea ice seasonality, with nearly sea ice-free conditions in summer in contrast with complete sea ice coverage in winter[3,4,17]. Moreover, the ESAS also directly receives a large volume of Russian pan-Arctic river discharge warming water, which controls the open water mass environment in summer[14,23,24] (Fig. 1b, c) and is thus an ideal region to assess the potential thermal impact of pan-Arctic rivers on Arctic sea ice. In summer, high levels of solar insolation increase the temperature of the river freshwater and also the volume of river runoff by intensifying the thawing of Siberian land snow/ice and permafrost, as well as producing more precipitation in the vast river drainage region[7,15,25,26]. As a result, ~74.5% (averaged in A.D. 1935–2013) of the annual suspended sediment is discharged from rivers into the ESAS during the early summer when the solar insolation is at a maximum[27], directly depending on both freshwater temperatures (affecting thermal erosion of Siberian ice-rich permafrost)[26,28,29] and river runoff amounts (impacting mechanical erosion by the transport capability of thawed and unconsolidated sediment)[23,30–32]. Since the pan-Arctic river thermal flux is jointly determined by river runoff and freshwater temperature[14,15]

(Supplementary Fig. 1), the sedimentation rate in the ESAS region since the MH, which is primarily controlled by the river material supply, acts as a potential proxy of the river thermal discharge, adding to the impact of sea level change.

In this study, after tracing the core sediment source during the Holocene by rare earth elements, we show the sand sediment fraction (>63 µm, %), known as the ice-rafted debris (IRD) proxy[33–35], and the sedimentation rate to reconstruct the Holocene variations in Arctic sea ice and regional river heat discharge, respectively (Methods, Supplementary Figs. 2–8). Based on high-resolution seismic profiles in the ESAS region (Supplementary Fig. 2), two gravity cores, LV77-36-1 (155.66°E, 74.10°N; water depth 36.0 m; recovery 376 cm; ~8.2 ka) and LV77-41-1 (154.12°E, 72.55°N; water depth 24.0 m; recovery 254 cm; ~7.0 ka), were collected (Fig. 1b, c). Their age-depth models and sedimentation rates were calculated based on 33 data points from radiocarbon (¹⁴C) dates and optically stimulated luminescence (OSL) ages of quartz after the application of polynomial analysis to assess the correlation of dating data between radiocarbon and OSL quartz ages and then the recalibration of OSL age data (Methods, Supplementary Figs. 3 and 4). Notably, although our cores are sufficiently long to show the history of the past 9000 years (Supplementary Figs. 5–9), we focused on the past 7500 years. During the MH (7.5–4.0 ka), the sea level was −0.75 ± 4.55 m, with few reconstruction points showing outliers up to ~−10 m, while it was 0.25 ± 2.45 m in the late Holocene (LH, 4.0–0 ka) relative to the present[36] (Fig. 2a). Because the paleo-coastal line has been relatively steady since the MH (Supplementary Fig. 10), this focus excluded the influence of changing distances between core locations and paleo-river mouths in the early Holocene.

Our measurements of marine proxies, together with the compilation of a published dataset for the sedimentation rate, sea ice, inland

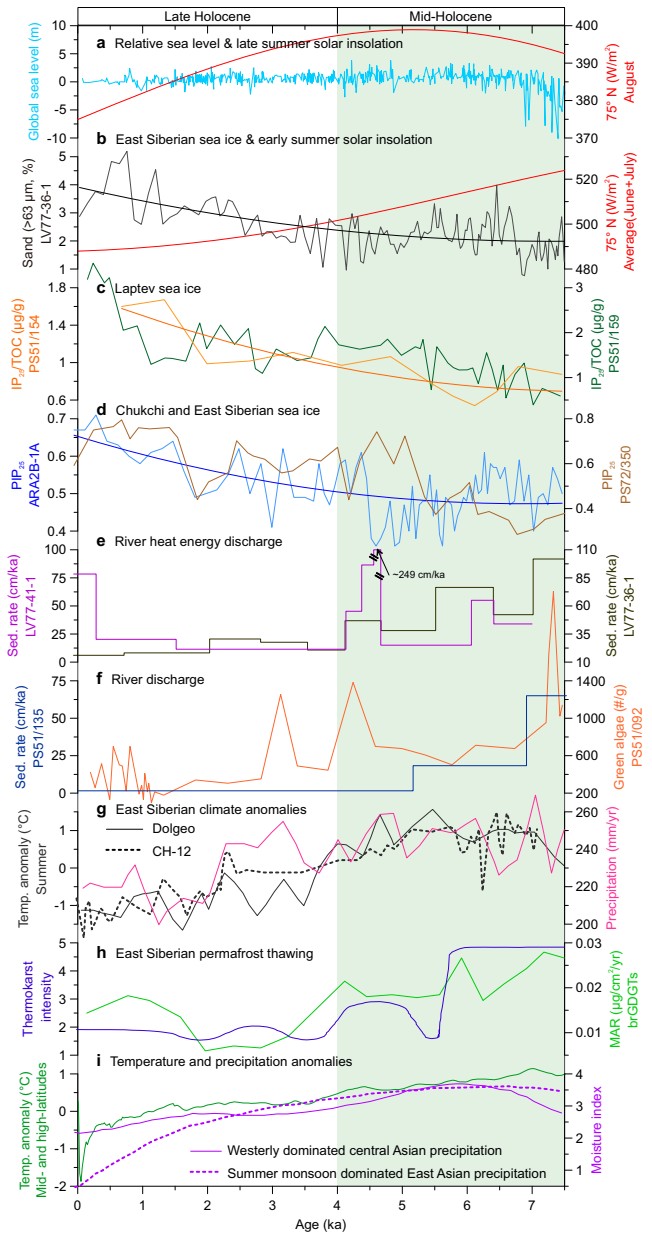

**Fig. 2 | Proxy records for the Arctic sea ice and pan-Arctic river environment during the Holocene. a** Global sea level[36] and solar insolation in late summer at 75° N[51]. **b–d** Solar insolation in early summer at 75°N[51] and reconstructions of sea ice change in the East Siberian Sea (this study), the Laptev Sea[18], and the Chukchi Sea[10]. The smooth lines represent the binomial expression of the data to obtain the Holocene orbital tendency. Reconstructions of Russian pan-Arctic river heat discharge in **e** the East Siberian Sea (this study) and **f** the Laptev Sea[49,63]. **g** Pollen-based reconstructions for the Russian Arctic temperature and precipitation in the summer season from the Dolgeo (solid black line)[22] and CH-12 (dotted black line) stations[20]. **h** Reconstruction of Siberian permafrost thawing[45,46]. **i** Summaries of air temperature in mid- and high-latitude regions[21] and the westerly dominated central (solid purple line) and summer monsoon dominated eastern Asian (dotted purple line) moisture in summer[19] based on hundreds of sedimentary records. The green band represents the mid-Holocene during 7.5–4.0 ka. Sed. Rate, MAR, and brGDGTs indicate the sedimentation rate, mass accumulation rate, and branched glycerol dialkyl glycerol tetraethers, respectively. See the reconstruction for the past 9000 years in Supplementary Fig. 9.

permafrost, basin surface air temperature, and precipitation, as well as observation data, reveal the coherent occurrences of the loss of Arctic sea ice and larger river heat discharges in the MH compared to the LH (Figs. 2 and 3). These results provide the important insight that the loss

of Arctic sea ice can be enhanced by the extensive pan-Arctic river heat discharge in early summer.

## Results

### ESAS sea ice in the Holocene
Our IRD (dominated by sea ice transport) record characterizes an increasing tendency from the MH to the LH (Fig. 2b). The relatively higher ESAS-sourced IRD during the LH compared to that during the MH is also recorded in both the Beaufort Sea[37] and Fram Strait[38], suggesting that the IRD was mainly governed by sea ice export from the ESAS. In contrast, the dinocyst-based reconstruction of sea ice in the marginal seas of Alaska has revealed a longer sea ice cover during the MH compared to the LH[39,40]. This result indicates that the external western Arctic sea ice could not influence the IRD records in the ESAS region, although it was transported by the Beaufort Gyre. Thus, all these IRD records indicate that the ESAS sea ice has continuously developed since the MH. Moreover, regional sea ice biomarker values gradually increased from the MH to the LH (Fig. 2b–d), suggesting relatively intensified sea ice melting in the MH spring and summer seasons[10,18]. In particular, micropaleontological records reveal that the ESAS experienced sea ice-free conditions in the MH summer[41]. Altogether, these results indicate that the regional sea ice significantly declined during the MH and increased during the LH.

### Russian pan-Arctic river heat discharge in the Holocene
In our results, the rare earth element data from the ESAS show minor oscillations around stable baseline values throughout the Holocene. This finding suggests a consistent source in which the stratified and flat shelf sediments were mainly from the adjacent Indigirka River (Fig. 1, Supplementary Figs. 2 and 7). Moreover, surface clay mineral data[34] and Holocene carbon isotopes and terrigenous biomarkers[42,43] also indicate that the sediment source in the shelf area was mainly the Siberian rivers, which coincides with the observation that the modern surface open water environment is dominated by river runoff (Fig. 1b, c). The calculated sedimentation rates and a synthesized dataset of the sedimentation rate in the river runoff-influenced ESAS region show higher sedimentation rates in the MH than in the LH, suggesting more discharge water input of the Russian pan-Arctic rivers into the study areas at this stage (Fig. 2e, f, Supplementary Fig. 8). This increased the total organic carbon in the shelf sedimentary budget during the MH[44], which also indicates the enhanced river water discharge (Supplementary Fig. 8b).

The larger Russian pan-Arctic river heat discharge was determined by both the increased river runoff and the higher freshwater temperature during the MH. Compared to the LH, the stronger summer solar insolation in the middle and high latitudes led to higher surface air temperatures by ~1–3 °C during the MH[20–22], resulting in intensified thawing of land snow/ice and permafrost (Fig. 2g–i). Previous biogeochemical studies have indicated that the insolation-driven warming condition severely deepened the thawed active layer of river basin (inland Siberia) ice-rich permafrost during the MH[45,46]. As the major transport route of thawed permafrost from inland to marginal seas, warming pan-Arctic rivers can further thaw permafrost and rapidly transport permafrost organic matter into the ESAS[31,47], and the amount of contributed terrigenous carbon during the MH was several times that during the LH[42,43]. Furthermore, climate modeling studies have indicated that warming can increase the global atmospheric moisture content via surface evaporation and synchronously reinforce poleward moisture transport from lower latitudes by an enlarged meridional moisture gradient[7,48] (Fig. 3). Thus, paleo-pollen records suggest that the regional precipitation increased in high-latitude Russia and mid-latitude Asia[20,22], and the moisture in the midlatitude watersheds upstream was transported by the polar vortex and East Asian summer monsoons from lower latitudes during the MH[19] (Fig. 2i, Supplementary Fig. 11). All these factors could have contributed to the high levels

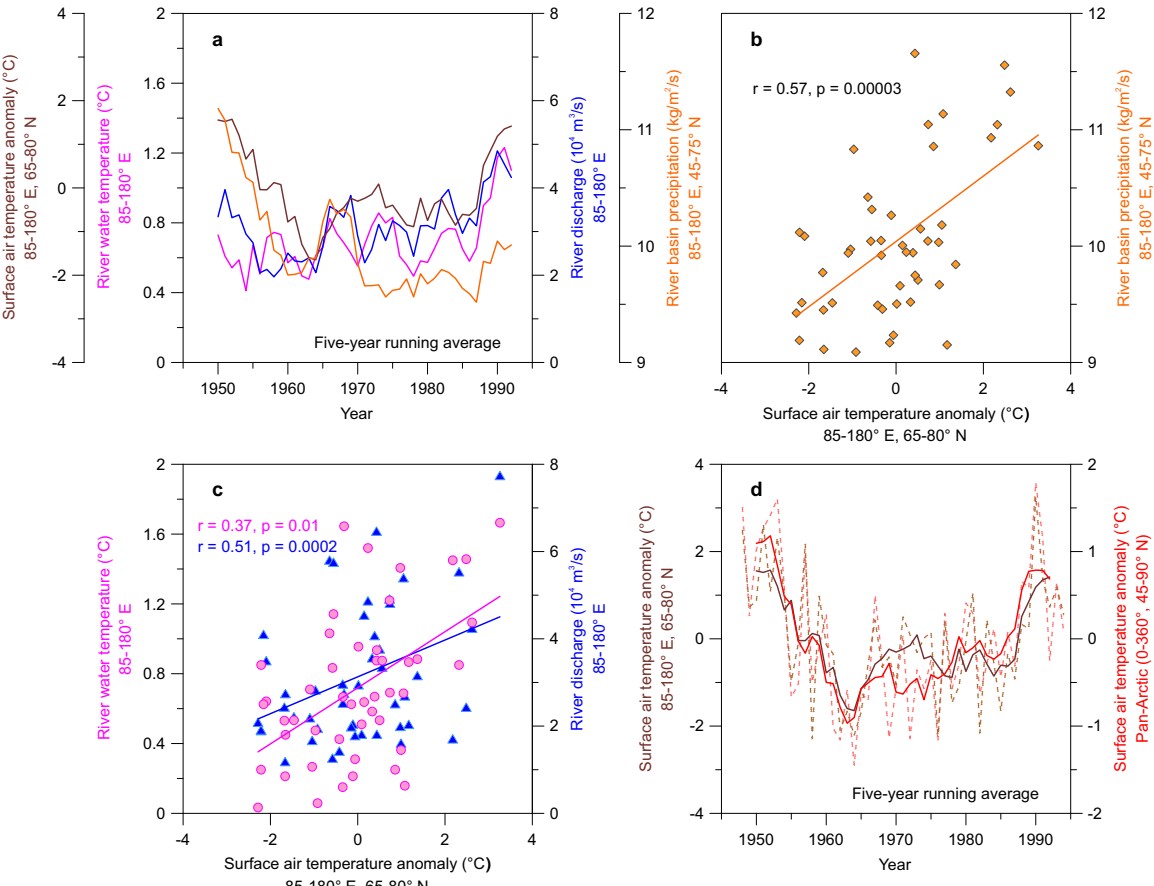

**Fig. 3 | Instrumental records of modern pan-Arctic river water discharge and temperature. a** Five-year running averages of the surface air temperature, river freshwater temperature, river runoff, and river basin precipitation in May from 1948 to 1994. **b** Surface air temperature in the East Siberian Arctic Shelf (ESAS) region versus river basin precipitation in May from 1948 to 1994. **c** Surface air temperature versus river freshwater temperature and surface air temperature versus river runoff. **d** Five-year running averages (solid lines) of the surface air temperatures in the ESAS area and over the pan-Arctic region from 1948 to 1994. It should be noted that spatial landform differences (e.g., lee side versus windward side) can result in considerable variability in precipitation across the pan-Arctic region[15], damming reduces river runoff, and permafrost thawing cools river freshwater[14]. However, all of the pan-Arctic basin-wide precipitation, river runoff, and water temperature peak in summer (Supplementary Fig. 1) and undergo an overall increase synchronously with global warming.

of river freshwater discharge and the large amounts of sediment transported into the ESAS during the MH. As a result, the salinity of the ESAS surface water decreased to ~13 PSU, and the Siberian Rivers discharged quantities of green algae and freshwater diatoms at this stage[49,50] (Fig. 2f).

Although the direct reconstruction of river water temperatures is lacking, station-based and instrumental measurements have suggested that the freshwater warming trend significantly and positively correlates with the surface air temperature based on ongoing global warming (Fig. 3). Accordingly, the multiple proxy-based reconstructions of higher surface summer temperatures[20–22] also imply higher river freshwater temperatures in high and middle latitudes during the MH. Therefore, these coherent increases in river runoff and river water temperatures indicate an intensified heat discharge from the surrounding Russian pan-Arctic rivers, contributing to higher regional sedimentation rates by increasing the sediment supply and transport during the MH.

## Discussion

In our records, the reconstructed sea ice growth since the MH coincides with the decreasing trend of solar insolation at 75° N averaged in early summer (June and July), while it is asynchronous with the late summer (August) solar insolation evolution with a maximum at ~5.0 ka[51] (Fig. 2a–d). This finding suggests that the early summer solar heat energy played an important role in determining the growth of

Arctic sea ice from the MH to the LH. This result is consistent with the indications based on a paleoclimate modeling study that the direct thermal impact of increased radiation has the potential to melt Arctic sea ice[6]. However, the loss of Arctic sea ice is attributed to direct solar insolation and albedo changes due to sea ice melting[6]. According to modern observations, the Arctic remains largely covered by sea ice in June and July[4], which restrains the direct absorption of solar insolation because of high albedos[16]. In comparison, our results reveal that there was a larger amount of runoff with warmer waters via Russian pan-Arctic rivers into the Arctic Ocean in the MH relative to the modern climate, thus enhancing the retreat of sea ice during the MH (Fig. 2e, f). At this point, we argue that the early summer solar insolation in boreal high latitudes acted to enhance the ESAS sea ice decline by increasing the runoff of warmer waters via pan-Arctic rivers along with the direct thermal impact of radiation during the MH.

Recent observations indicate that the heat discharge of Russian pan-Arctic rivers peaks in June and July in response to the maximum of the early summer solar insolation (Supplementary Fig. 1). Thus, during the MH, the intensified river freshwater runoff and river heat input could have directly melted the early summer Arctic sea ice northward as far as >80° N through advection under floes and then warm the open surface water (Figs. 1b, c, 2a–h). This condition is similar to the present situation in that the early summer river heat discharge results in the loss of sea ice in the shelf areas[13] and then drives the maximum open sea surface temperature to exceed 8 °C in mid-June[12] and 12 °C at the

beginning of July[11]. As a positive feedback mechanism in response to the enhanced river heat input during the MH, the significant loss of Arctic sea ice along with sediment- and phytoplankton-rich freshwater could have prominently reduced the early summer sea ice albedo during the MH (Fig. 2b–f), absorbing an additional ~64% of the incident summer solar heat energy[13] because the early summer solar insolation is much larger than that in late summer[51]. This could further increase the open-sea surface temperature and melt the regional sea ice, leading to the expansion of seasonal ice-free regions and periods in the MH in early summer. During the later summer, the increasing heat storage in the marginal seas, which originated from river discharge and solar insolation, would continuously promote the development of this positive feedback and subsequently reduce the thickness of the sea ice during the MH. Thus, the earlier melting of thinner sea ice in early summer could have cumulatively intensified the effect of river heat discharge on the loss of Arctic sea ice and further increased the early summer absorption of solar insolation throughout the MH. Given that the river runoff-dominated shelves are important source areas of Arctic seasonal sea ice[33,34,52], the increased river heat discharge provided an important preconditioning melt of sea ice over the Arctic Ocean in spring and early summer during the MH.

In addition, different factors have regulated Holocene Arctic sea ice changes. Although the Atlantic water inflow at intermediate water depths does not impact the surface sea ice through a direct hydrodynamic process (Fig. 1a), the heat contained in it and subsurface Pacific water can be released upward to the sea surface[8–10]. However, except for the direct thermal impact of rivers, the larger freshwater mass of pan-Arctic rivers has also led to low surface salinities that stabilize the summer water column, thereby inhibiting upward heat release from the deeper layers into the surface ocean[53], which would act against the mechanisms proposed here. In contrast, modern observations show that the upward heat release from warm and salty Atlantic water inflows peaks in winter[8], whereas the warm and low-salinity Pacific water inflow intensifies during August–October[54]. Thus, the Atlantic and Pacific water inflows act to melt the Arctic sea ice later than the control by the river heat discharge in early summer. Throughout the Holocene, biotic and biomarker records indicate that the Atlantic water inflow exhibited a relatively stable state with millennial-scale variability since the MH[55–57], also similar to the Pacific water inflow[58,59] (Supplementary Fig. 12). The relatively comparable inflows between the MH and the LH further highlight the roles of higher solar insolation and river heat discharge in reducing sea ice during the MH. In addition, the low concentrations of atmospheric methane[60] and carbon dioxide[61] also contrasted with the loss of Arctic sea ice during the MH.

Overall, our study suggests that the increased Russian pan-Arctic river heat discharge contributed to the melting of early summer Arctic sea ice in the ESAS area as an indirect consequence of the higher solar insolation during the MH summer (Fig. 4). Our findings for the ESAS area may also explain the retreat of sea ice over the Arctic shelf seas during the MH, but spatial heterogeneity according to the relationship between the loss of sea ice and pan-Arctic river heat discharge is likely given ongoing global warming[13–15]. Thus, the basin-scale impact of the pan-Arctic river heat discharge on the sea ice under the warmer climate conditions during the MH is a topic for future studies. Moreover, based on paleoclimate records, our findings indicate the feedback of permafrost thawing, melting, and warm runoff in conjunction with the decline of sea ice in the Arctic Ocean. This mechanism is not well presented in most climate models and is also a topic for further developments in modeling.

## Methods
### Materials and age model
During cruise LV77 using the vessel R/V "Akademik Lavrentiev" in August and September 2016, two sediment cores, LV77-36-1 and LV77-41-1, were obtained on the ESAS and offshore of the mouth of the

Indigirka River, respectively (Fig. 1b, c, Supplementary Fig. 2). Both core sediments were mainly composed of ash-black and dark-grayish (80% black) clayey silt without obviously incorporated sand laminations or bioturbation structures (Supplementary Fig. 3). Bivalve shells (≤4 mm) were sporadically distributed in core LV77-36-1, while they were nearly absent in core LV77-41-1. The similar sedimentary facies from top to bottom indicate that the sedimentary environment did not significantly differ. Based on observations of the sea surface salinity and temperature, core station LV77-41-1 was persistently under the influence of water discharged from the Indigirka River in both warm and cold seasons (Supplementary Fig. 13). Thus, core LV77-41-1 provided a valid position to study the Holocene changes in the internal Indigirka River. In comparison, the influence of river runoff on the sediments in core LV77-36-1 was relatively weakened, and thus, these sediments provided continuous records to study the geological changes on the ESAS.

The age model of sediment core LV77-36-1 is based on twelve accelerator mass spectrometry [14]C dating analyses of bivalve shells and seven OSL dating analyses of quartz, while the age-depth model of core sediment LV77-41-1 is based on three radiocarbon dating analyses of bivalve shells and ten OSL dating analyses of quartz (Supplementary Figs. 3 and 4 and Supplementary Tables 1 and 2). The radiocarbon data were measured by Beta Analytic (Miami). Each [14]C age was calibrated to calendar years before A.D. 1950 via Calib 8.2 software. The referenced dataset was Marine 13.14 C[62]. The delta R (local reservoir age) was set as $53 \pm 67$ years, which was calculated based on five age points collected in the Laptev Sea[63]. Between two adjacent calendar age points, the interpolated ages and 2-sigma uncertainties were calculated using the Clam age-depth estimate program 2.2 via open-source R statistical software[64]. Additionally, all the age-depth models of the referenced ESAS sediment cores based on the [14]C dating analyses of shells were recalibrated using the same dataset and statistical method to compare the various proxies.

Before subsampling, black plastic cylindrical boxes (H 5 cm × D 3 cm) were used to obtain the OSL dating samples, which were then immediately sealed with covers and black tape and wrapped with an opaque plastic bag to avoid light exposure. The samples were pretreated and measured in the optical dating laboratory with red LED illumination at the Qinghai Institute of Salt Lakes, Chinese Academy of Sciences (Xining). Because the sufficiently bleached quartz in the ≥63 µm fraction during sea ice transport[65,66] was not abundant enough to be analyzed during this interglacial period, the quartz samples in the 4–11 µm fraction[67], which were referenced by the grain-size distributions (Supplementary Fig. 5), were alternatively measured using an automated Risø TL/OSL reader model DA-20 equipped with blue ($\lambda = 470 \pm 20$ nm) and infrared radiation diodes ($\lambda = 830$ nm). Blue LEDs at 130 °C for 40 s and a U-340 filter (thickness 7.5 mm, detection window 275–390 nm) in advance of the photomultiplier tube were used to stimulate and detect the luminescence, respectively. The U and Th concentrations of the bulk samples were measured using thermal iCAP RQ inductively coupled plasma–mass spectrometry (ICP–MS), and the K concentration was analyzed using a thermal iCAP7400 inductively coupled plasma optical emission spectrometer (ICP–OES). In addition, the wet and dry samples were weighed to calculate the water content, and an error of ±5% was estimated to reconstruct the paleowater content of the sediment. The detailed processes of the sample pretreatment, measurement, and subsequent analyses followed those of ref. 68 and ref. 69.

In the study area, because the core sediment in the 4–11 µm fraction is mainly sourced from the Siberian Rivers[34,42,43], the sedimentary processes from the source to the sink can prevent quartz from being exposed to light. First, inland East Siberia (sediment source region) has been covered with permafrost and land snow/ice for the long term[45,70]. Second, a large mass of sediment due to river erosion is rapidly transported into the marginal seas in certain months (mainly in June and July[27,31,32]), resulting in high sedimentation rates on the ESAS[63,71]. Hence, the older age estimations by OSL dating were

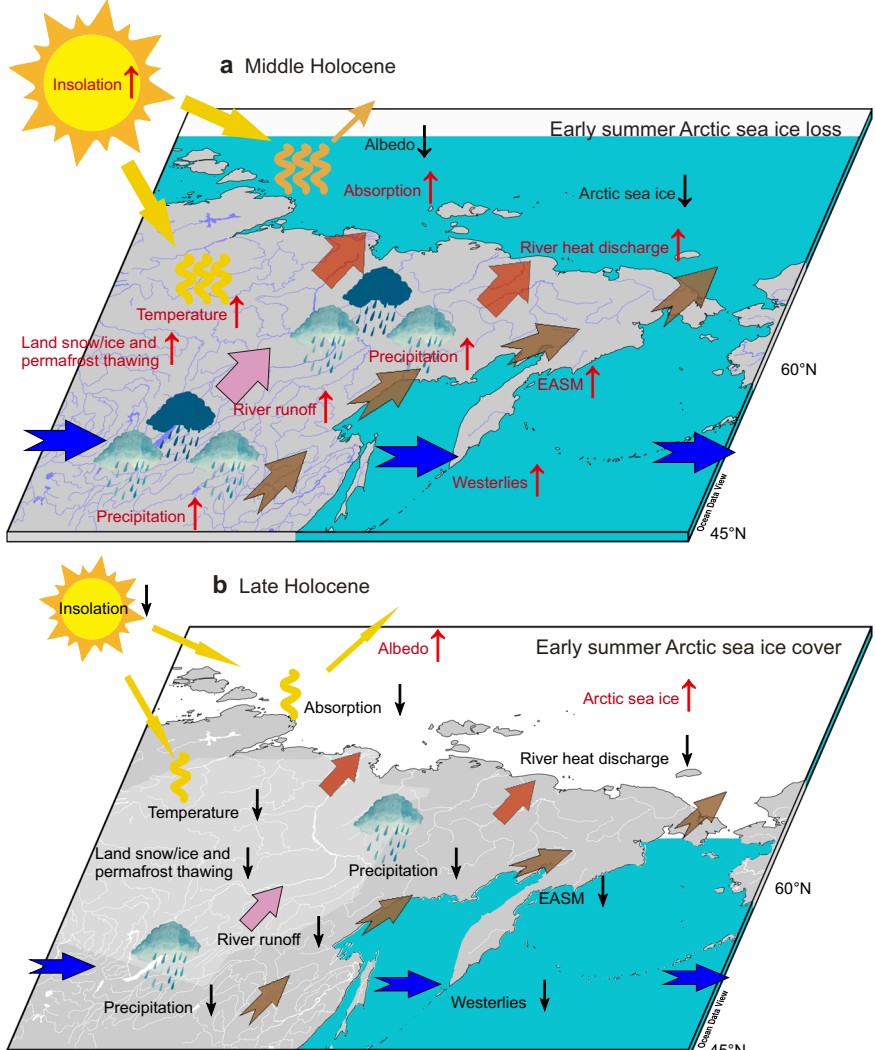

**Fig. 4 | Mechanisms of Russian pan-Arctic river heat discharges on Arctic sea ice changes. a** During the mid-Holocene, the intensified early summer solar insolation resulted in mid- and high-latitude Asian positive temperature anomalies and enhanced the river runoff that originated from reinforced permafrost, land snow/ice thawing, and river basin precipitation. Meanwhile, the increased precipitation and associated heat content were mainly from the lower latitudes due to the enhanced meridional moisture gradient, which could have been transported by the intensified East Asian summer monsoon (EASM) and the polar vortex, including the westerlies. As a result, the increased river heat discharge enhanced the loss of Arctic sea ice in early summer and strengthened the absorption of solar energy inputs. **b** During the late Holocene, the opposite conditions prevailed. In the Arctic Ocean, the blue areas represent the loss of sea ice in early summer, while the white regions point to sea ice cover in early summer.

observed along with the higher sedimentation rate during the Holocene (Supplementary Fig. 4). Finally, after river-dominated sediment deposition, long-term sea ice cover results in high sea ice/snow albedo[4,16]. In contrast, the absence of an erosional surface and the smooth changes in the grain size show that the core sediment is not significantly reworked by ocean currents (Supplementary Fig. 3a). In contrast, the difference values between the OSL age and the ¹⁴C ages have decreased since 9.0 ka, and the values were much larger than the transport times from the river mouth to the core location (~1.5 ka)[72], particularly in the MH (Supplementary Fig. 4). Thus, inadequate light exposure to river-discharged quartz in the 4–11 μm fraction leads to age overestimations by OSL dating[65–67,73–75].

In this study, a high-confidence polynomial equation was employed to recalibrate the OSL dating of quartz to calendar years and to compare it with the calibrated ¹⁴C ages from both sediment cores because the OSL ages are linearly correlated with the radiocarbon age throughout the Holocene (Supplementary Fig. 3). The maximum difference between the (re)calibrated OSL date and the ¹⁴C age was 0.6 ka,

while the average value was 0. This result can effectively reduce their differences, and thus, the sedimentary records based on these age models can be directly compared between the MH and the LH. The resulting age-depth models reveal similar trends of decreasing sedimentation rates since ~8.2 ka. The mean values of the sedimentation rate were 63.1 cm/ka (core LV77-36-1) and 75.4 cm/ka (core LV77-41-1) during the MH (7.5–4.0 ka), which contrasts with values of 22.7 cm/ka (core LV77-36-1) and 36.7 cm/ka (core LV77-41-1) during the LH (Supplementary Fig. 3).

In addition, the published data on the sedimentation rate on the ESAS were also referenced and compared with our data. These sedimentation rate records[50,63,71], which present the MH and LH differences, are presented in Supplementary Fig. 8. In comparison, sediment cores, which are out of the region of influence of summer pan-Arctic river-discharged freshwater, were not considered in this study because the sediment cores were mainly located in the deep seas (e.g., Lomonosov Ridge) and the Chukchi Sea, where the sediments originated from multiple sources, such as long-distance transport from the Kara

and Barents Seas via Atlantic Water inflow, Bering Sea via Pacific Water inflow, and river discharge, although the sedimentation rate decreased from the MH to the LH[10,41]. This data compilation reveals that the average sedimentation rate in the MH was higher than that in the LH, which is in line with our sedimentation rate (Supplementary Fig. 8).

## Grain size

The grain sizes of terrigenous sediment were measured using a Mastersizer 3000 laser diffraction particle size analyzer with a measurement range of 0.01–3,500 µm. In the laboratory of the First Institute of Oceanography (Qingdao), 299 samples in total were collected at 2-cm intervals after being described and photographed. Before the instrument mensuration, each bulk sample was pretreated with hydrogen peroxide (15 ml, 15%) for 24 h at 20 °C and then placed in a thermostat water bath (85 °C) for 2 h. This process was repeated thrice to remove all organic matter. After removing the biogenic carbonate fraction by pretreating it with hydrochloric acid (5 ml, 3 mol/L) for 24 h at 20 °C, the detrital sediment and biogenic silica were isolated. Subsequently, these residual samples were rinsed with distilled water (3×) to make the PH values of the solution equal to ~7. The samples were then treated with sodium carbonate solution (20 ml, 2 mol/L) for 4 h at 85 °C in a thermostat water bath to remove the fraction of biogenic silica. Later, the residual materials were rinsed with distilled water (3×) to reduce the influence of reagents; then, potential aggregates of samples were disaggregated using an ultrasonic cleaner for 1 min. Finally, grain-size measurements were immediately performed in deionized and degassed water to avoid gaseous influence. Each measurement was repeated twice to ensure that the results were repeatable (Supplementary Fig. 5). In addition, sortable silt parameters were also analyzed based on ref. 76 to discuss the influence of the Siberian Coastal Current on the sedimentary records (Supplementary Fig. 6).

## Rare earth elements

The measurements of rare earth elements (La, Ce, Pr, Nd, Pm, Sm, Ru, Gd, Tb, Dy, Ho, Re, Tm, Yb, and Lu) of the bulk samples were performed using a Thermal iCAP RQ ICP–MS system (Supplementary Fig. 7). In the laboratory of the First Institute of Oceanography (Qingdao), 117 samples in total (80 samples from core LV77-36-1 with a temporal resolution of ~100 years and 37 samples from core LV77-41-1 with a temporal resolution of ~200 years) were freeze-dried, ground, and then directly digested by a concentrated $HF + HNO_3$ mixture in Teflon vessels. Blank samples and national standard substance GBW07309 were used to monitor and test the procedures, and one out of every ten samples was randomly selected to repeat the test. The errors of replication were less than 5%.

## Observation data

Previous studies have suggested that recent global warming has caused the daily maximum discharge to occur earlier in spring (from June to late May)[70,77]. Thus, the surface air temperature anomalies, river water temperatures, and river runoff data in May from A.D. 1948 to 1994 were linearly analyzed to reveal the evolutionary relationship among them. The observation data of the surface air temperature anomalies and Siberian River basin precipitation were from NCEP reanalysis data[78] (https://psl.noaa.gov/), while the station-based data of Russian pan-Arctic river runoff and freshwater temperature were from R-ArcticNet (www.r-arcticnet.sr.unh.edu/). The data for the ESAS (85–180°E, 65–80°N) and pan-Arctic (0–360°E, 45–90°N) surface air temperature anomalies, as well as the Siberian River basin (85–180°E, 45–75°N) precipitation, were regional average values. The river water discharge data were summed, while the river freshwater temperature (May 5–June 5) data were averaged for the Yenisey, Lena, Yana, Indigirka, and Kolyma Rivers. These representative river basins are located in Siberia (85–180°E), and they are nearly completely covered by permafrost[14]. The datasets for the gauging stations included Igarka

(Yenisey River), Kusur (Lena River), Ubileynaya and Djahgky (Yana River), Vorontsovo (Indigirka River), and Kolymskoye and Srednekolymsk (Kolyma River). The river runoff datasets for the Yenisey, Lena, and Indigirka Rivers were complete from A.D. 1938 to 1994. Values for the Yana River between A.D. 1972 and 1989 were averaged between datasets from the Ubileynaya (dataset from A.D. 1972 to 1994) and Djahgky (dataset from A.D. 1938 to 1989) gauging stations because they are adjacent. Values for the Kolyma River at the Kolymskoye gauging station (dataset from A.D. 1978 to 1994) from 1938 to 1977 were derived from the next upstream gauging station at Srednekolymsk (dataset from A.D. 1938 to 1988)[79], according to the linear equation[79] between these stations for an overlapping period of 11 years ($r = 0.82$, $p = 0.002$). Because the NCEP reanalysis datasets of the surface air temperature were complete from A.D. 1948 to the present, the overlapping period (A.D. 1948–1994) of river runoff, water temperature, and air temperature was analyzed in this study.

## Data availability

Source data in this study are available at https://doi.org/10.17632/pp6ntt6549.3.

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

## Acknowledgements

We are grateful to Dr. V.N. Karnaukh for providing data on high-resolution seismic profiles. Dr. Zhi Dong helped with data collection. Dr. Xin Zhou helped with OSL sampling and pretreatment. Mrs. Mao Yuan and Dr. Jingjing Gao helped with the grain size and rare earth element measurements, respectively. This study was funded by the National Natural Science Foundation of China (42130412) to X.S., the Ministry of Science and Technology of the People's Republic of China (2019YFE0125000) to X.G., the National Natural Science Foundation of China (42006199 and 42076074) to J.D. and L.H., respectively, the Natural Science Foundation of Shandong Province, China (ZR2020QD110) to J.D., the National Social Science Foundation of China (19ZDA140) to A.A., the Marine S&T Fund of Shandong Province for Pilot National Laboratory for Marine Science and Technology (Qingdao) (2018SDKJ0104-3) to X.S., and the Taishan Scholar Program of Shandong (tspd 20181216) to X.S. The expeditions were partly supported by the Ministry of Sciences and Education of the Russian Federation (Project No. AAAA-A17-117030110033-0).

## Author contributions

J.D. initiated this study and performed lab measurements and data analysis. X.S. organized the cruise and sample collection, designed the study progress, and obtained funding. X.G. interpreted the data. A.A. led the cruise and jointly collected the samples. L.H., X.L., G.Y., Y.W., Y.V., S.Q., A.B., and G.L. contributed to the data analysis. J.D., X.S., and X.G. drafted the paper. J.D., X.S., X.G., A.A., L.H., X.L., G.Y., Y.W., Y.V., S.Q., A.B., and G.L. contributed to the refinement and revision of the paper.

## Competing interests

The authors declare no competing interests.
