## [Peer Review File · Nature Communications]

Enhanced Arctic sea ice melting controlled by larger heat discharge of mid-Holocene riversReviewers' Comments:

Reviewer #1:

Remarks to the Author:

This paper presents evidence to support increase heat flux by Siberian rivers during the mid-Holocene that may have caused increased sea ice and permafrost melt with implications for global warming today. The new data presented consists primarily of IRD abundance for the past 9ka, and mass accumulation rates.

The work is of significance to the field of paleoclimatology and has implications for present day global warming.

Because the emphasis is on the mid-Holocene, this should be in the paper's title. The paper could be shortened by moving all discussion of seismic stratigraphy to the supplemental materials because this has little to do with the focus of the paper. The paper could also benefit by previous published data on IRD increases at 3-7ka from the Laptev and Kara Seas and IRD peaks at 2, 5, and 9.5 ka from the E. Siberian Sea (Indigirka and Kolyma Rivers) based on detrital Fe grain geochemical fingerprinting (Darby et al., 2002; Darby, 2012).

The paper could also benefit from the detailed textural data from actual sea ice sediment in Darby et al. (2009). I was also surprised to see no references to the work of deVernal on the extent of sea ice during the Holocene (deVernal et al., 2013; 2005).

Line 5: The use of overlooked in reference to sea ice loss during the Holocene due to river input here is a bit misleading (Refs 38, 44). A better term would be understudied.

Line 36: A better term for signally would be significantly.

Line 99-102: Here mention is made of a gradual increase in IRD and thus ice-rafting and presumed ice production between 7ka and the present day. Yet actual IRD data show rapid increases or peaks in Fe grains from Circum-Arctic sources during this time (Darby et al., 2012).

Lines 173-181: The discussion of deeper Atlantic Water here seems to contradict earlier statement that this water is too deep to affect the shallow E. Siberian Sea water. So why the need to establish a "fresh" water halocline to protect against Atlantic water convection?

Line 212: Core LV77-41-1 is not located in the Indigirka River estuary but off shore of the river's mouth.

Line 268: A clearer term for well-diversified would be disaggregated.

Line 270: The term mensuration refers to mathematical calculations and a better term here would be to use analysis or measurement.

Fig. 3: The correlation between T degC and precipitation is less than convincing. Perhaps a brief discussion as to why there is so much spread in the data would be warranted.

References

Darby, D.A., Bischof, J., Spielhagen, R., Marshall, S., and Herman, S., 2002, Arctic ice export events and their potential impact on global climate during the late Pleistocene. *Paleoceanography*, 17(2): 15.1-15.17.

Darby, D.A., Ortiz, J. Polyak, L., Lund, S., Jakobsson, M., and Woodgate, R.A., 2009. The role of currents and sea ice in both slowly deposited central Arctic and rapidly deposited Chukchi-Alaskan margin sediments. *Global Planet. Change*. 68: 58-72. doi:10.1016/j.gloplacha.2009.02.007

Darby, D. A., Ortiz, J. D., Grosch, C., and Lund, S., 2012, 1,500 year cycle in the Arctic Oscillation identified in Holocene Arctic sea-ice drift. *Nature Geoscience*, doi: 10.1038/NGEO1629. doi: 10.1038/NGEO1629.

Dennis Darby

Reviewer #2:

Remarks to the Author:

In this paper the authors attempt to illustrate that enhanced heating of arctic rivers in the mid-Holocene led to enhanced sea ice retreat across the broad East Siberian Arctic Shelf. They emphasize that this is an important mechanism, pertinent to modern and future Arctic change, that remains somewhat overlooked. I think the idea is compelling, and the authors have done an impressive job of pulling together diverse data-sets from across the Siberian shelf that partially support their assertion. However, I am left with a few reservations about publishing the manuscript in its current form. I believe that most/all of these can be readily fixed and would encourage a re-submission of a revised manuscript.

1) On the surface, an apparent weakness of the paper was the rather basic sedimentological proxies that are used to infer warming/discharge of river waters and sea ice retreat, namely that 1) that sedimentation rates across the shallow shelf are directly proportional to river discharge, and 2) the coarse fraction content in their record is an accurate predictor of sea ice conditions.

However, the authors do present more sophisticated proxies from other published records from the region that (to a first order) seem to support these assertions (i.e biomarker based sea-ice reconstructions, and GDGT flux rates – as a proxy for permafrost derived organic matter). This important 'data-synthesis' aspect of the paper does not feel complete or adequately explained in the text. The sedimentation rate and grain size measurements from the two new marine sediment cores are not enough to make this a compelling new contribution – the paper relies on the synthesis of data sets that is presented in Fig. 2. As such I believe -

a) A stronger emphasis needs to be made/placed on synthesizing a large number of data sets. This aspect of the paper needs to be brought forward in the abstract. Furthermore, the synthesized data-sets need to be more thoroughly explained in the main text, and not simply referenced to other published studies in the figure captions.

b) More records need to be compiled and analysed. There are a lot of published sedimentary records from this region of the Arctic shelf. If sedimentation rates are a robust indicator of river discharge/inland temperatures, than I think this should be shown by compiling and illustrating more records. To some extent this has been done in papers that are not referenced by the authors – for example: Wegner, C., et al., 2015. Variability in transport of terrigenous material on the shelves and the deep Arctic Ocean during the Holocene. *Polar Research*, 34, 24964. doi: 10.3402/polar.v34.24964 The purpose of compiling more records on Holocene sedimentation rates should be to more robustly illustrate that this is a regionally significant trend. This is not really done in the current paper – but seems entirely possible.

2) I think the sedimentological analysis and presentation of the new data from LV77-41-1 and LV77-36-1 is underdeveloped. Specifically,

a) Since the grain size data is generated using a laser particle analyser, it should be relatively straightforward to look at indices for current sorting- namely sortable silt mean size and % sortable silt of the fines. It is not clear how much of the high frequency and long-term variability in the grain size record originates from changes in the oceanographic current conditions along the shelf. This may

or may not challenge the idea that the grain size is directly related to sea ice. For example: McCave, I. and Andrews, J., 2019. Distinguishing current effects in sediments delivered to the ocean by ice. I. Principles, methods and examples. *Quaternary Science Reviews* 212 (2019) 92-107

b) Prior to analysis, the authors treated their grain size samples to remove biogenic carbonate. This is good, but I am concerned that there is no mention of the concentration or potential influence of biogenic silica – which I suspect is much more abundant on the shallow shelf. Could this be biasing the grain size records, or contributing to the elevated coarse fraction contents towards the modern? Are they really presenting a terrigenous grain size record as they report, or do significant contributions from biogenic silica remain?

c) There is no mention of what enhanced erosion of inland permafrost would contribute to the sedimentary budget. Many studies have focused on the enhanced contribution and biogeochemical signatures of organic matter following the deglacial transgression of the shelf – but these are not discussed and few of them are mentioned. Does the organic carbon content of the sediments also increase during periods of enhanced river discharge? Has this been measured on these cores? Even if it shows little change, it would be an important supplementary data set that could more fully describe the sedimentology of these new records.

d) Abundances of Rare Earth Elements (REE) are used to show that the source of the sediments for the middle and late Holocene have remained constant, and are derived from East Siberian Rivers. I think in this context it is important to show the downcore changes in abundance of the REE and not just the average Holocene values to more thoroughly illustrate that the major sediment sources have not changed. Furthermore, they suggest on Line 106 that this interpretation of their REE data is supported by 'clay mineral data' published by Nurnberg et al. (1994) – but it is left unclear how? This is an example of the authors needing to provide more clarity in the text about what they are comparing and reporting.

3) The OSL dates are an important part of the age modelling. The authors have been a bit negligent here when it comes to justifying, discussing and reporting the data. This needs to be corrected in a revised manuscript. For example, they attribute the age offset between the OSL and 14C dates to incomplete bleaching that commonly occurs during sea ice transport. However, they have not referenced or discussed any of the findings by authors who have studied and applied OSL dating of quartz grains in marine sediments from the Arctic – for example:

Berger, G. W. (2006). Trans-arctic-ocean tests of fine-silt luminescence sediment dating provide a basis for an additional geochronometer for this region. *Quaternary Science Reviews*, 23.

Berger, G. W. (2009). Zeroing tests of luminescence sediment dating in the Arctic Ocean: Review and new results from Alaska-margin core tops and central-ocean dirty sea ice. *Global and Planetary Change*, 68(1–2), 48–57. <https://doi.org/10.1016/j.gloplacha.2009.03.019>

Berger, G. W. (2011). Surmounting luminescence age overestimation in Alaska-margin Arctic Ocean sediments by use of 'micro-hole' quartz dating. *Quaternary Science Reviews*, 30(13–14), 1750–1769. <https://doi.org/10.1016/j.quascirev.2011.03.019>

Berger, G. W., & Polyak, L. (2012). Testing the use of quartz 'micro-hole' photon-simulated luminescence for dating sediments from the central Lomonosov Ridge, Arctic Ocean. *Quaternary Geochronology*, 11, 42–51. <https://doi.org/10.1016/j.quageo.2012.04.008>

Jakobsson, M., Backman, J., Murray, A., & Løvlie, R. (2003). Optically Stimulated Luminescence dating supports central Arctic Ocean cm-scale sedimentation rates. *Geochemistry, Geophysics, Geosystems*, 4(2). <https://doi.org/10.1029/2002GC000423>

West, G., Alexanderson, H., Jakobsson, M., and O'Regan, M., (2021). Optically stimulated luminescence dating supports pre-Eemian age for glacial ice on the Lomonosov Ridge off the East Siberian continental shelf. *Quaternary Science Reviews*, 267,

<https://doi.org/10.1016/j.quascirev.2021.107082>

All of these authors find that quartz is sufficiently bleached during sea ice transport, and Berger in particular has found that the fine size fractions are less reliable than the coarser size fractions – particularly because they can be transported and reworked by ocean currents. Transport of the 4-11 micron size fraction used in this study cannot be solely attributed to top-down deposition from sea ice. Maybe the offsets are related to cross-shelf transport times, which (for organic matter) have been shown to be very large across the Laptev shelf:

Bröder, L., Tesi, T., Andersson, A. et al. Bounding cross-shelf transport time and degradation in Siberian-Arctic land-ocean carbon transfer. *Nat Commun* 9, 806 (2018).
<https://doi.org/10.1038/s41467-018-03192-1>

4.) The authors have not made reference to important paleoclimate modelling works that have formerly concluded that Holocene river discharge of Eurasian rivers has been increasing since the middle Holocene.

Wagner et al., 2011. Arctic river discharge trends since 7 ka BP. *Global and Planetary Change*. 79(1–2), p. 48-60. <https://doi.org/10.1016/j.gloplacha.2011.07.006>

And to a lesser extent the review of –

Wegner, C., et al., 2015. Variability in transport of terrigenous material on the shelves and the deep Arctic Ocean during the Holocene. *Polar Research*, 34, 24964. doi: 10.3402/polar.v34.24964

5) In revising the manuscript the final points of the discussion also need to be more developed – specifically on clarifying the relative role that warming/increased river run-off may have on Arctic sea ice. What are the basin-scale influences? Is it really more important than Pacific water inflow, atlantification or the direct insolation response?

Statements like the following:

“At this point, we argue that the early summer solar insolation in boreal high latitudes has controlled the ESAS sea ice growth since the Holocene by synchronously increasing runoff of warmer waters via pan-Arctic rivers rather than by the direct thermal impact of radiation.”

Are rather vague, not adequately referenced and unsupported by the existing/presented data.

For example should this be discussed in relation to studies such as this:

Stranne et al., 2014. Arctic Ocean perennial sea ice breakdown during the Early Holocene Insolation Maximum. *Quaternary Science Reviews*, 92, 123-132,
<https://doi.org/10.1016/j.quascirev.2013.10.022>

In summary, I do believe that this manuscript has the potential to be a very valuable contribution, but in its current form I feel the arguments are not fully developed and/or supported by the presented data. I do think the authors can rectify this given the opportunity. This will require adding some additional data as well as re-structuring and re-writing large parts of the text – as such my recommendation is for a major revision.

Response to Reviewer #1:

This paper presents evidence to support increase heat flux by Siberian rivers during the mid-Holocene that may have caused increased sea ice and permafrost melt with implications for global warming today. The new data presented consists primarily of IRD abundance for the past 9ka, and mass accumulation rates.

The work is of significance to the field of paleoclimatology and has implications for present day global warming.

Thanks for the helpful suggestions for our manuscript. In this revision work, we've made a reply to each comment and also made the corresponding change in the revised manuscript.

1) Because the emphasis is on the mid-Holocene, this should be in the paper's title. The paper could be shortened by moving all discussion of seismic stratigraphy to the supplemental materials because this has little to do with the focus of the paper. The paper could also benefit by previous published data on IRD increases at 3-7ka from the Laptev and Kara Seas and IRD peaks at 2, 5, and 9.5 ka from the E. Siberian Sea (Indigirka and Kolyma Rivers) based on detrital Fe grain geochemical fingerprinting (Darby et al., 2002; Darby, 2012).

The paper could also benefit from the detailed textural data from actual sea ice sediment in Darby et al. (2009). I was also surprised to see no references to the work of deVernal on the extent of sea ice during the Holocene (deVernal et al., 2013; 2005).

References

Darby, D.A., Bischof, J., Spielhagen, R., Marshall, S., and Herman, S., 2002, Arctic ice export events and their potential impact on global climate during the late Pleistocene.

Paleoceanography, 17(2): 15.1-15.17.

Darby, D.A., Ortiz, J. Polyak, L., Lund, S., Jakobsson, M., and Woodgate, R.A., 2009. The role of currents and sea ice in both slowly deposited central Arctic and rapidly deposited Chukchi-Alaskan margin sediments. Global Planet. Change. 68: 58-72. doi:10.1016/j.gloplacha.2009.02.007

Darby, D. A., Ortiz, J. D., Grosch, C., and Lund, S., 2012, 1,500 year cycle in the Arctic Oscillation identified in Holocene Arctic sea-ice drift. Nature Geoscience, doi: 10.1038/NGEO1629.

Following this comment, we've changed the title to 'Enhanced Arctic sea ice melting controlled by larger heat discharge of Mid-Holocene rivers'. In addition, we've moved the discussion about seismic stratigraphy to the Supplementary Information, please see lines 166-173 in the revised Supplementary Information.

Moreover, thanks for reminding us about the published works of Darby et al. (2002, 2009, 2012) and de Vernal et al. (2013, 2015). Indeed, their datasets are supportive of our discussion in this work. In the revised manuscript, we've added the data of Fe grain geochemical fingerprinting (Darby et al., 2002; Darby et al., 2012) to support the declined regional sea ice during the mid-Holocene compared to the late Holocene conditions, please see the lines 101-103 in revised Main Text and lines 91-100 in revised Supplementary Information. In parallel, the textual data of actual sea ice sediment in Darby et al. (2009) is also supportive evidence to our illustration about the sea-ice influence on the parameters in sediment grain size, please see lines 85-87 and 206-209 in revised Supplementary Information. Furthermore, the findings of de Vernal et al. (2005, 2013) are in line with our findings that the western Arctic sea ice, which was carried by the Beaufort gyre, could not influence the Holocene sea ice change in the East Siberian Sea, please see lines 103-106 in the revised Main Text. Besides, we've found one extra reference that de Vernal et al. (2020) also show biomarker proxy evidence to support the significant Arctic sea ice loss in the mid-Holocene compared to late Holocene, please see lines 110-111 in the revised Main Text.

2) *Line 5: The use of overlooked in reference to sea ice loss during the Holocene due to river input here is a bit misleading (Refs 38, 44). A better term would be understudied.*

Thanks for correcting us, and we've changed 'overlooked' to 'unresolved', please see lines 23-24 in the revised Main Text.

3) *Line 36: A better term for signally would be significantly.*

Thanks for correcting us, and we've changed this sentence to "As indicated by the modern global warming process, a stronger thermal flux associated with discharge water by pan-Arctic rivers would have contributed to the MH summer sea ice loss due to the higher-than-present solar insolation".

Please see lines 53-56 in the revised Main Text.

4) Line 99-102: Here mention is made of a gradual increase in IRD and thus ice-rafting and presumed ice production between 7ka and the present day. Yet actual IRD data show rapid increases or peaks in Fe grains from Circum-Arctic sources during this time (Darby et al., 2012).

We agree with the reviewer that the IRD and Fe grains records show distinct patterns during the increase from the mid-Holocene to late Holocene. Here, the IRD and Fe grain records have commonly suggested the Arctic sea ice expansion during the late Holocene compared to the mid-Holocene conditions. On the other hand, we attribute the distinct evolution patterns between the IRD and Fe grains to their inconsistent drift paths (Darby et al., 2012). In the sampling area, a major factor that is responsible for the IRD abundance is suspension freezing with rising frazil ice (Darby et al., 2011; Nürnberg et al., 1994; Reimnitz et al., 1993), and the impact of external sea ice advection on the regional IRD records is limited (de Vernal et al., 2005; de Vernal et al., 2013). In contrast, the Fe grains are affected by not only sea ice export from the sampling area but also advective sea ice from the outside (Darby et al., 2012). Therefore, the Fe grains records represent the combined signal of the gradual change in the sea ice export and the rapid shift in the Arctic Oscillation (Darby et al., 2012). Please see lines 91-100 in the revised Supplementary Information.

5) Lines 173-181: The discussion of deeper Atlantic Water here seems to contradict earlier statement that this water is too deep to affect the shallow E. Siberian Sea water. So why the need to establish a “fresh” water halocline to protect against Atlantic water convection?

We agree with the reviewer that our previous illustration is not clear. First, the relatively warmer river discharged water thermally melts sea ice in summer (Park et al., 2020). Second, the lower salinities of the river discharged water inhibits the relatively warmer water from the deeper layer into the surface ocean by increasing a saline stratification (Prange and Lohmann, 2003). At this point, the low-salinity characteristic of river discharged water plays a sort of side-effect role in restraining the upwelling thermal impact of the Atlantic subsurface water in melting the regional sea ice. Please see lines 198-204 in the revised Main Text and lines 76-78 in the revised Supplementary Information.

6) Line 212: Core LV77-41-1 is not located in the Indigirka River estuary but off shore of the river's mouth.

As mentioned by the reviewer, core LV77-41-1 allocates offshore of the Indigirka River mouth. Based on the observations of sea surface salinity and temperature in May and September, the core location LV77-41-1 is persistently under the impact of river discharged water in both warm and cold seasons (Supplementary Fig. 12). Thus, LV77-41-1 is a valid position to study the impact of the Indigirka River on the East Siberian Arctic Shelf region. Please see lines 237-242 in the revised Main Text and newly added Supplementary Fig. 12.

Newly added supplementary Fig. 12 Sea surface salinity and temperature in the study area. **a, c** The sea surface salinity in the cold (May) and warm (September) months, respectively. **b, d** The sea surface temperature in the cold (May) and warm (September) months, respectively. Data are sourced from World Ocean Atlas 2013 (Locarnini et al., 2013; Zweng et al., 2013).

7) Line 268: A clearer term for well-diversified would be disaggregated.

Thanks. We've replaced 'well-diversified' with 'disaggregated', please see lines 327-328 in the revised Main Text.

8) *Line 270: The term mensuration refers to mathematical calculations and a better term here would be to use analysis or measurement.*

Thanks. We've corrected 'mensuration' to 'measurement', please see lines 330-331 in the revised Main Text.

9) *Fig. 3: The correlation between T degC and precipitation is less than convincing. Perhaps a brief discussion as to why there is so much spread in the data would be warranted.*

Following this comment, we've added a discussion of the correlation between surface air temperature and precipitation. Observation and simulation have commonly suggested that the recent warming causes an increase in the global atmospheric moisture content via surface evaporation and also synchronously reinforces poleward moisture transport from lower latitudes by enlarged meridional moisture gradient (Bintanja et al., 2020; Zhang et al., 2012). This thus determines the more Russian pan-Arctic river basin precipitation that positively correlates with present surface air warming, although the perturbation by the precipitation variability due to regional landforms (e.g., lee side versus windward side) (Yang et al., 2021). Please see lines 137-140 and 620-624 in the revised Main Text and revised Supplementary Fig. 1b.

Response to Reviewer #2:

In this paper the authors attempt to illustrate that enhanced heating of arctic rivers in the mid-Holocene led to enhanced sea ice retreat across the broad East Siberian Arctic Shelf. They emphasize that this is an important mechanism, pertinent to modern and future Arctic change, that remains somewhat overlooked. I think the idea is compelling, and the authors have done an impressive job of pulling together diverse data-sets from across the Siberian shelf that partially support their assertion. However, I am left with a few reservations about publishing the manuscript in its current form. I believe that most/all of these can be readily fixed and would encourage a re-submission of a revised manuscript.

Thanks for the constructive suggestions on our manuscript, and we've made a reply to every comment and also revised the manuscript accordingly.

1) On the surface, an apparent weakness of the paper was the rather basic sedimentological proxies that are used to infer warming/discharge of river waters and sea ice retreat, namely that 1) that sedimentation rates across the shallow shelf are directly proportional to river discharge, and 2) the coarse fraction content in their record is an accurate predictor of sea ice conditions.

However, the authors do present more sophisticated proxies from other published records from the region that (to a first order) seem to support these assertions (i.e biomarker based sea-ice reconstructions, and GDGT flux rates – as a proxy for permafrost derived organic matter). This important 'data-synthesis' aspect of the paper does not feel complete or adequately explained in the text. The sedimentation rate and grain size measurements from the two new marine sediment cores are not enough to make this a compelling new contribution – the paper relies on the synthesis of data sets that is presented in Fig. 2. As such I believe -

a) A stronger emphasis needs to be made/placed on synthesizing a large number of data sets. This aspect of the paper needs to be brought forward in the abstract. Furthermore, the synthesized data-sets need to be more thoroughly explained in the main text, and not simply referenced to other published studies in the figure captions.

Following this comment, we've added the systemization of our new measurement with published

datasets, and also further discussion about the enhanced Arctic sea ice melting due to larger heat discharge by mid-Holocene rivers. Hence, we've made a change in the abstract that the Holocene reconstructions of Arctic sea ice and Russian pan-Arctic river heat discharge are based on our records of ice-rafted debris (IRD) and sedimentation rates, together with a compilation of published paleoclimate and observation data. Please see lines 24-27 in the revised Main Text.

Moreover, we've added a new data compilation for the published sedimentation rates in the river runoff-dominated East Siberian Arctic Shelf region (Bauch et al., 2001; Bauch and Polyakova, 2003; Wegner et al., 2015), sea ice (Hörner et al., 2016; Stein et al., 2017), and also extra paleoclimate records including IRD (Darby et al., 2002; Darby et al., 2012), biomarkers (de Vernal et al., 2005; de Vernal et al., 2020; de Vernal et al., 2013), inland permafrost (Morgenstern et al., 2013; Winterfeld et al., 2018), precipitation (Chen et al., 2019; Klemm et al., 2016; Klemm et al., 2013), water and air temperature (Klemm et al., 2016; Krumpfen et al., 2013; Marcott et al., 2013), etc., to clearer present the thermal impact of the Russian pan-Arctic rivers on the Arctic sea ice loss. Here, the new data compilation is supportive of our proposed mechanism that the Enhanced Arctic sea ice melting was controlled by larger heat discharge of Mid-Holocene rivers. Please see lines 100-156 in the revised Main Text, lines 48-152 in the revised Supplementary Information, and new references.

More specifically, the explanations associated with the new data compilation mainly include the following three aspects:

(i) Both our IRD and the East Siberia-sourced Fe grains records (Darby et al., 2002; Darby et al., 2012) show an increase from the mid-Holocene to late Holocene, commonly suggesting the Arctic sea ice loss during the mid-Holocene. Similarly, the biomarker proxies of IP₂₅ and Dinocysts records also indicate the regional sea ice decline in the East Siberian Sea during the mid-Holocene (de Vernal et al., 2020; Hörner et al., 2016; Stein et al., 2017). Please see lines 100-112 in the revised Main Text.

(ii) Both our sedimentation rate records and a synthesized dataset of the sedimentation rates in the river runoff-influenced East Siberian Arctic Shelf region (Bauch et al., 2001; Bauch and Polyakova, 2003; Wegner et al., 2015) commonly show higher regional sedimentation rates in the mid-Holocene, compared to the late Holocene conditions. This suggests more discharge water input of the Russian pan-Arctic rivers into the study areas during the mid-Holocene. Please see lines 115-126 in the revised Main Text and newly added Supplementary Fig. 8.

(iii) The pan-Arctic river thermal flux is jointly determined by river runoff and the freshwater temperature (Lammers et al., 2007; Yang et al., 2021). In the new data compilation, biomarkers records (e.g., organic flux, $\Delta^{14}\text{C}$, $\delta^{13}\text{C}$, brGDGTs, lignin) suggest an intensified thawing of the permafrost in the river basin area in the mid-Holocene (Morgenstern et al., 2013; Winterfeld et al., 2018), while the paleo pollen records indicate the coherently enhanced precipitation (Chen et al., 2019; Klemm et al., 2016; Klemm et al., 2013). These records commonly suggest larger river runoff discharge in the mid-Holocene. In parallel, higher river freshwater temperatures are inferred by warmer mid-Holocene air temperatures based on an observation-based relationship that the river freshwater temperature increases synchronously with the surface air temperatures (Klemm et al., 2016; Krumpfen et al., 2013; Marcott et al., 2013). Overall, the paleoclimate data for the increased river runoff discharge and the higher river freshwater temperatures commonly suggest the larger thermal discharge by the Russian pan-Arctic rivers into the Arctic Ocean during the mid-Holocene. Please see lines 128-156 in the revised Main Text, and **also see the reply to the following comment by the reviewer.**

b) More records need to be compiled and analysed. There are a lot of published sedimentary records from this region of the Arctic shelf. If sedimentation rates are a robust indicator of river discharge/inland temperatures, than I think this should be shown by compiling and illustrating more records. To some extent this has been done in papers that are not referenced by the authors – for example:

Wegner, C., et al., 2015. Variability in transport of terrigenous material on the shelves and the deep Arctic Ocean during the Holocene. Polar Research, 34, 24964. doi: 10.3402/polar.v34.24964.

The purpose of compiling more records on Holocene sedimentation rates should be to more robustly illustrate that this is a regionally significant trend. This is not really done in the current paper – but seems entirely possible.

Following this comment, we've added a synthesis of the published sedimentation rates (Bauch et al., 2001; Bauch and Polyakova, 2003; Wegner et al., 2015) in the river runoff-dominated East Siberian Arctic Shelf region. As shown in Supplementary Fig. 8, the averaged sedimentation rate in the mid-Holocene is higher than the late Holocene condition, in both our measurement and the published

data collection. Please see lines 121-125 and 303-313 in the revised Main Text and newly added Supplementary Fig. 8. In addition, based on the new data compilation, the paleoclimate records for the increased river runoff discharge and the higher river freshwater temperatures commonly indicate the larger thermal flux by the Russian pan-Arctic rivers into the Arctic Ocean in the mid-Holocene. This result is in line with the indication of the regional sedimentation rate. Please see lines 128-156 in the revised Main Text. Please see further information in our reply to Question 1a in this comment.

Newly added Supplementary Fig. 8 Data compilation of the published sedimentation rates in the East Siberian Arctic Shelf region. **a** Sea surface salinity (PSU) and core locations. **b-h** The sedimentation rate records from the cores of (b) LV77-36-1, (c) LV77-41-1, (d) PS51/159, (e) PS51/135, (f)

PS51/092, (g) PM9462, and (h) PS51/080 (Bauch et al., 2001; Bauch and Polyakova, 2003; Wegner et al., 2015). The green bands show the mid-Holocene period of 7.5 to 4.0 ka. In (b), the purple curve exhibits the total organic carbon (Astakhov et al., 2019). In addition, the Box-Whisker plots show the median (middle dotted line), 25th and 75th percentile (box), and 5th and 95th percentile (whiskers) as well as outliers (single points).

2) *I think the sedimentological analysis and presentation of the new data from LV77-41-1 and LV77-36-1 is underdeveloped. Specifically,*

*a) Since the grain size data is generated using a laser particle analyser, it should be relatively straightforward to look at indices for current sorting- namely sortable silt mean size and % sortable silt of the fines. It is not clear how much of the high frequency and long-term variability in the grain size record originates from changes in the oceanographic current conditions along the shelf. This may or may not challenge the idea that the grain size is directly related to sea ice. For example: McCave, I. and Andrews, J., 2019. Distinguishing current effects in sediments delivered to the ocean by ice. I. Principles, methods and examples. *Quaternary Science Reviews* 212 (2019) 92-107*

Following this comment, the sortable silt mean size and the percentage from cores LV77-36-1 and LV77-41-1 have been calculated and analyzed. As shown in Supplementary Fig. 6, the sortable silt mean size at core LV77-36-1 increased from the mid-Holocene to the late Holocene. However, the grain size sorting has become poor with increasing IRD (sand) percentage since the mid-Holocene (Supplementary Fig. 5), which contrasts with the possibly intensifying hydrodynamic transport by the Siberian Coastal Current. Moreover, a nonsignificant correlation between sortable silt mean size and the percentage suggests that the sortable silt is not sufficiently well current-sorted to provide a reliable flow history during the Holocene (McCave and Andrews, 2019). In particular, a previous study has suggested that sortable silt can also be transported by sea ice, based on the grain-size distribution of actual sea ice sediment (Darby et al., 2009). Therefore, we argue that the positive correlation between sortable silt mean size and IRD records in core LV77-36-1 is linked to the impact of sea ice on the regional IRD record. Please see lines 79-89 in the revised Supplementary Information and the newly added Supplementary Fig. 6.

Newly added Supplementary Fig. 6 Sortable silt records in the Holocene. **a–b** The sortable silt mean size and percentage in cores LV77-36-1 and LV77-41-1. **c** The sortable silt percentage versus mean size in core LV77-36-1, and the IRD versus sortable silt mean size. **d** The sortable silt percentage versus mean size in core LV77-41-1.

b) Prior to analysis, the authors treated their grain size samples to remove biogenic carbonate. This is good, but I am concerned that there is no mention of the concentration or potential influence of biogenic silica – which I suspect is much more abundant on the shallow shelf. Could this be biasing the grain size records, or contributing to the elevated coarse fraction contents towards the modern? Are they really presenting a terrigenous grain size record as they report, or do significant contributions from biogenic silica remain?

Following this comment, we've measured additionally new 299 samples of grain size to evaluate the influence of biogenic silica via pretreating with sodium carbonate solution (Na_2CO_3 , 20 ml, 2 mol/L) for 4 hours at 85 °C in a thermostat water bath. The new measurement results indicate that biogenic silica insignificantly changes the grain size distributions of core sediment (Supplementary Fig. 5) because very few diatoms were observed in the sediment deposited in the study area during the

Holocene (Bauch and Polyakova, 2003). Moreover, during the late Holocene, the richer IRD is in line with the higher values in the East Siberia-sourced Fe grains in the cores from the Alaskan marginal sea (Darby et al., 2012) and the Fram Strait (Darby et al., 2002). Thus, the coarse fraction (sand) presents a terrigenous grain size record, and it can be used as an IRD proxy. Please see lines 316-333 in the revised Main Text and revised Supplementary Fig. 5.

Revised Supplementary Fig. 5 The grain size records of **a** LV77-36-1 and **b** LV77-41-1.

c) There is no mention of what enhanced erosion of inland permafrost would contribute to the sedimentary budget. Many studies have focused on the enhanced contribution and biogeochemical signatures of organic matter following the deglacial transgression of the shelf – but these are not discussed and few of them are mentioned. Does the organic carbon content of the sediments also increase during periods of enhanced river discharge? Has this been measured on these cores? Even if it shows little change, it would be an important supplementary data set that could more fully describe the sedimentology of these new records.

Following this comment, we've additionally involved the total organic carbon records in the revised analysis, which has been published (Astakhov et al., 2019), please see Supplementary Fig. 8b. As shown, the organic carbon content becomes higher coherent with the increased river discharge in the mid-Holocene. This is likely attributed to a larger budget from the river basin permafrost thawing to the sedimentary sinks via the Siberian Rivers during the mid-Holocene, due to stronger summer solar insolation at this stage (Mann et al., 2015; Martens et al., 2020; Morgenstern et al., 2013; Tesi et al., 2016). Please see lines 125-137 in the revised Main Text and the newly added Supplementary Fig. 8b (also shown on Page 13 in this letter).

d) Abundances of Rare Earth Elements (REE) are used to show that the source of the sediments for the middle and late Holocene have remained constant, and are derived from East Siberian Rivers. I think in this context it is important to show the downcore changes in abundance of the REE and not just the average Holocene values to more thoroughly illustrate that the major sediment sources have not changed. Furthermore, they suggest on Line 106 that this interpretation of their REE data is supported by 'clay mineral data' published by Nurnberg et al. (1994) – but it is left unclear how? This is an example of the authors needing to provide more clarity in the text about what they are comparing and reporting.

Following this comment, we've shown the downcore changes in the abundance of the rare earth elements. As shown in Supplementary Fig. 7, the rare earth elements records stay stable with only oscillations throughout the Holocene in the cores of both LV77-36-1 and LV77-41-1, suggesting a consistent sediment source to the study area. Please see lines 115-118 in the revised Main Text and revised Supplementary Fig. 7.

In addition, we've revised the text to provide more clarity about a comparison between our data with the previous proxy records. For instance, we've rediscussed the REE and clay mineral data by Nurnberg et al. (1994). In our results, the rare earth element data from the ESAS show only oscillations based on a relatively consistent level throughout the Holocene. This suggests a consistent source that the stratified and flat shelf sediments are mainly from the adjacent Indigirka River, which is one of the Siberian rivers (Supplementary Figs. 2 and 7). Moreover, surface clay mineral data (Nürnberg et al., 1994) and Holocene carbon isotopes and terrigenous biomarkers (Martens et al., 2020; Tesi et al., 2016)

also indicate that the sediment source is mainly the Siberian rivers, which coincides with the observation that the open surface water environment in the East Siberian Arctic Shelf region is dominated by river runoff. Thus, Holocene sediment was mainly sourced from the Siberian River discharged material. Please see lines 118-121 in the revised Main Text.

Revised Supplementary Fig. 7 Abundance of rare earth elements in core sediments during the Holocene. **a** LV77-36-1. **b** LV77-41-1. **c** The upper continental crust (UCC) composite-normalized rare earth elements of the river suspension (Rachold, 1999) and in the core sediment. The normalized values are that the average of each rare earth element during the given time phase (e.g., 2.0–0 ka) divides the related value in the UCC (Taylor and McLennan, 1995). The distribution modes in core

LV77-36-1 are similar to those in core LV77-41-1, suggesting that the core sediment source was mainly from the adjacent Indigirka River throughout the Holocene.

3) *The OSL dates are an important part of the age modelling. The authors have been a bit negligent here when it comes to justifying, discussing and reporting the data. This needs to be corrected in a revised manuscript. For example, they attribute the age offset between the OSL and 14C dates to incomplete bleaching that commonly occurs during sea ice transport. However, they have not referenced or discussed any of the findings by authors who have studied and applied OSL dating of quartz grains in marine sediments from the Arctic – for example:*

*Berger, G. W. (2006). Trans-arctic-ocean tests of fine-silt luminescence sediment dating provide a basis for an additional geochronometer for this region. *Quaternary Science Reviews*, 23.*

*Berger, G. W. (2009). Zeroing tests of luminescence sediment dating in the Arctic Ocean: Review and new results from Alaska-margin core tops and central-ocean dirty sea ice. *Global and Planetary Change*, 68(1–2), 48–57. <https://doi.org/10.1016/j.gloplacha.2009.03.019>*

*Berger, G. W. (2011). Surmounting luminescence age overestimation in Alaska-margin Arctic Ocean sediments by use of ‘micro-hole’ quartz dating. *Quaternary Science Reviews*, 30(13–14), 1750–1769. <https://doi.org/10.1016/j.quascirev.2011.03.019>*

*Berger, G. W., & Polyak, L. (2012). Testing the use of quartz ‘micro-hole’ photon-simulated luminescence for dating sediments from the central Lomonosov Ridge, Arctic Ocean. *Quaternary Geochronology*, 11, 42–51. <https://doi.org/10.1016/j.quageo.2012.04.008>*

*Jakobsson, M., Backman, J., Murray, A., & Løvlie, R. (2003). Optically Stimulated Luminescence dating supports central Arctic Ocean cm-scale sedimentation rates. *Geochemistry, Geophysics, Geosystems*, 4(2). <https://doi.org/10.1029/2002GC000423>*

*West, G., Alexanderson, H., Jakobsson, M., and O’Regan, M., (2021). Optically stimulated luminescence dating supports pre-Eemian age for glacial ice on the Lomonosov Ridge off the East Siberian continental shelf. *Quaternary Science Reviews*, 267, <https://doi.org/10.1016/j.quascirev.2021.107082>*

All of these authors find that quartz is sufficiently bleached during sea ice transport, and Berger in

particular has found that the fine size fractions are less reliable than the coarser size fractions – particularly because they can be transported and reworked by ocean currents. Transport of the 4–11 micron size fraction used in this study cannot be solely attributed to top-down deposition from sea ice. Maybe the offsets are related to cross-shelf transport times, which (for organic matter) have been shown to be very large across the Laptev shelf:

Bröder, L., Tesi, T., Andersson, A. et al. Bounding cross-shelf transport time and degradation in Siberian-Arctic land-ocean carbon transfer. Nat Commun 9, 806 (2018). <https://doi.org/10.1038/s41467-018-03192-1>

Following this comment, we've made a double check about the OSL dating. In this study, we've only compared the sedimentary records between the mid-Holocene and the late Holocene. We confirm that the age model based on the recalibrated OSL data of quartz in the 4–11 μm fraction can achieve this aim. Thus, we've not changed the age model, which has been reconstructed in the earlier version of the manuscript (Supplementary Fig. 3). By referencing the recommended articles (Berger, 2006, 2009, 2011; Berger and Polyak, 2012; Broder et al., 2018; Jakobsson et al., 2003; West et al., 2021), we've provided the detailed reasons for the material selection, the age overestimations, and the recalibrated results of OSL dating of the quartz in the 4–11 μm fraction.

(i) Because the sufficiently bleached quartz in the $\geq 63 \mu\text{m}$ fraction during sea-ice transport (Berger and Polyak, 2012; West et al., 2021) is not abundant enough to be analyzed during the Holocene, the quartz samples in the 4–11 μm fraction (Berger, 2011), which were referenced by the grain-size distributions (Supplementary Fig. 5), were alternatively measured in this study. Please see **lines 261–266 in the revised Main Text**.

(ii) Because the sediment in the 4–11 μm fraction in the study area is mainly sourced from the Siberian Rivers (Martens et al., 2020; Nürnberg et al., 1994; Tesi et al., 2016), the sedimentary process from the source to the sink can prevent light exposure of sediment. First, the river basin (sediment source area) is long-term covered with permafrost and land snow/ice (Karlsson et al., 2016; Winterfeld et al., 2018; Yang et al., 2002). Second, a large mass of sediment due to river erosion is rapidly transported into the marginal seas in limited months (mainly during June–July) (Costard et al., 2014; Magritsky et al., 2017; Tananaev, 2016), resulting in regionally high sedimentation rates (Bauch et al., 2001; Wegner et al., 2015). Hence, the older age estimations by OSL dating occurred along with the higher sedimentation

rate during the Holocene (Supplementary Fig. 4). Finally, after river-dominated sediment deposition, the long-term sea ice cover results in high albedo (Comiso et al., 2008; Perovich et al., 2007). In contrast, the absence of an erosional surface and the smooth changes in grain size show that the core sediment is not significantly reworked by ocean currents. On the other hand, the difference values between OSL age and ^{14}C ages have decreased since 9.0 ka, and the values were much larger than the transport times from the river mouth to the core location (~ 1.5 ka) (Broder et al., 2018), particularly in the mid-Holocene (Supplementary Fig. 4). Therefore, the age overestimations by OSL dating of quartz in the 4–11 μm fraction are mainly attributed to the inadequate light exposure to the river discharged quartz. Please see lines 276-289 in the revised Main Text and newly added Supplementary Fig. 4.

(iii) Because the OSL ages linearly correlate with the radiocarbon age, a high-confidence polynomial equation was employed to recalibrate the OSL dating of quartz to calendar years and compare it with the calibrated ^{14}C ages from both sediment cores (Supplementary Fig. 3). The maximum difference between the recalibrated OSL date and ^{14}C age was 0.6 ka, while the average value was 0. This result can effectively reduce their differences, and thus, the sedimentary records based on these age models can be directly compared between the mid-Holocene and the late Holocene. Please see lines 291-301 in the revised Main Text and Supplementary Fig. 3.

Newly added Supplementary Fig. 4 The influence of sedimentation on OSL age. **a** Difference values between OSL age and accelerator mass spectrometry (AMS) ^{14}C age in core LV77-36-1. **b** Sedimentation rate in core LV77-36-1.

4.) *The authors have not made reference to important paleoclimate modelling works that have formerly concluded that Holocene river discharge of Eurasian rivers has been increasing since the middle Holocene.*

Wagner et al., 2011. Arctic river discharge trends since 7 ka BP. *Global and Planetary Change*. 79(1–2), p. 48-60. <https://doi.org/10.1016/j.gloplacha.2011.07.006>

And to a lesser extent the review of –

Wegner, C., et al., 2015. Variability in transport of terrigenous material on the shelves and the deep Arctic Ocean during the Holocene. *Polar Research*, 34, 24964. doi: 10.3402/polar.v34.24964

Thanks for the reminding, and we've involved the discussion about the two studies in the revised manuscript. Using the coupled atmosphere-ocean circulation model ECHO-G, Wagner et al. (2011) have suggested a slight increase in the Eurasian river discharges from the mid-Holocene to the late Holocene, followed by a pronounced intensification during the Pre-industrial era. This is caused by a more rapid decrease in local net evaporation compared to a decline in moisture due to a relative cooling trend since the mid-Holocene. On the other hand, the river basin permafrost is another key factor in changing the terrestrial hydrological cycle but is not considered in their modeling results due to the missing of a permafrost module (Wagner et al., 2011). At the end of this study, Wagner et al. (2011) have encouraged more discussion about the impact of permafrost on the pan-Arctic rivers, as a future perspective. Here, based on paleoclimate records, we suggest that the permafrost thawing process plays an important role in enhancing the river discharge of the Russian pan-Arctic in the mid-Holocene, and overcoming the evaporation change by air temperatures. In the study by Wegner, et al., 2015, they have compiled a pan-Arctic review about the changing Holocene sources, transport, and sinks of terrigenous sediment in the Arctic Ocean. In their results, the sedimentation rates over the Arctic shelves are greater in the Early Holocene until ~7.0 ka due to a rapid sea-level rise, then the sedimentation rates show a declining trend from 7.0 ka to 0 ka. Thus, the compilation of the published sedimentation rate is in line with our sedimentation rate data, suggesting a decreased river discharge since the mid-Holocene (Supplementary Fig. 8). Please see lines 137-152 in the revised Supplementary Information, and newly added Supplementary Fig. 8.

5) *In revising the manuscript the final points of the discussion also need to be more developed –*

specifically on clarifying the relative role that warming/increased river run-off may have on Arctic sea ice. What are the basin-scale influences? Is it really more important than Pacific water inflow, atlantification or the direct insolation response?

Statements like the following:

“At this point, we argue that the early summer solar insolation in boreal high latitudes has controlled the ESAS sea ice growth since the Holocene by synchronously increasing runoff of warmer waters via pan-Arctic rivers rather than by the direct thermal impact of radiation.”

Are rather vague, not adequately referenced and unsupported by the existing/presented data.

For example should this be discussed in relation to studies such as this:

Stranne et al., 2014. Arctic Ocean perennial sea ice breakdown during the Early Holocene Insolation Maximum. Quaternary Science Reviews, 92, 123-132, <https://doi.org/10.1016/j.quascirev.2013.10.022>

Based on the simulations by a coupled atmosphere-sea ice-ocean model, Stranne et al. (2014) have suggested that higher solar insolation acts to melt sea ice via a directly radioactive effect. Physically, such direct impact by higher solar insolation on sea ice melting is inevitable, no matter background climate conditions. Therefore, we agree with the reviewer that our previous state ‘*rather than by the direct thermal impact of radiation*’ is too conclusive. More accurately, our findings of the indirect impact of the higher solar insolation on sea ice melting via increasing river heat discharge are to make up a missing issue in the whole process of the pronounced mid-Holocene sea ice loss.

On the other hand, the atlantification in the modern Arctic Ocean occurs coherently with the rapid sea ice melting along with the ongoing warming (Polyakov et al., 2017). In contrast, the Atlantic water inflow shows a relatively stable state with millennial-scale variability since the mid-Holocene (Berben et al., 2014; Giraudeau et al., 2010), also similar to the Pacific water inflow (Harada et al., 2014; Ruan et al., 2017). Here, both the relatively comparable inflows between the mid-Holocene and late Holocene highlight the roles of higher solar insolation in reducing the sea ice during the mid-Holocene. Notably, based on paleoceanographic records, it is a challenge to quantitatively conclude whether the direct control (via radioactive impact) or indirect control (via the river heat discharge) plays a more important role. Thus, we argue that the higher river heat discharge and directly more radioactive impact coherently contribute to the enhanced sea ice loss in the mid-Holocene summer. In the revised

manuscript, we've changed this sentence to “*At this point, we argue that the early summer solar insolation in boreal high latitudes acts to enhance the ESAS sea ice decline by increasing runoff of warmer waters via pan-Arctic rivers along with the direct thermal impact of radiation during the MH*”.

Please see lines 159-214 in the revised Main Text.

In the entire Arctic Ocean, studies have shown the spatial heterogeneity in the distribution of pan-Arctic river heat discharge (Lammers et al., 2007; Yang et al., 2021) and the regional sea ice retreat (Comiso et al., 2008; Lindsay and Schweiger, 2015) in the ongoing global warming. These studies likely suggest the similar existence of spatial heterogeneity in the control of higher river heat discharge in reducing sea ice based on the mid-Holocene conditions. Here, we suggest a data compilation for the entire pan-Arctic basin to evaluate the basin-scale effect of the river heat discharge as a future perspective depending on data availability. Please see lines 216-226 in the revised Main Text.

In summary, I do believe that this manuscript has the potential to be a very valuable contribution, but in its current form I feel the arguments are not fully developed and/or supported by the presented data. I do think the authors can rectify this given the opportunity. This will require adding some additional data as well as re-structuring and re-writing large parts of the text – as such my recommendation in for a major revision.

Thanks again for all the constructive comments. In particular, our major revision work has newly remeasured 299 grain size samples and added a data compilation of the published proxy records, including sedimentation rates in the river runoff-dominated shelf area, IRD, biomarkers, permafrost, precipitation, water and air temperature, etc. All the datasets have been carefully reanalyzed and presented to indicate the thermal impact of the Russian pan-Arctic rivers on the Arctic sea ice loss.

References (Newly involved references in the revised manuscript are shown in blue, with the black ones used in the response to the comments in this letter):

Astakhov, A.S., Sattarova, V.V., Shi, X.F., Hu, L., Aksentov, K.I., Alatortsev, A.V., Kolesnik, O.N., Mariash, A.A., 2019. Distribution and sources of rare earth elements in sediments of the Chukchi and East Siberian Seas. *Polar Science* 20, 148-159.

Bauch, H.A., Mueller-Lupp, T., Taldenkova, E., Spielhagen, R.F., Kassens, H., Grootes, P.M., Thiede, J., Heinemeier, J., Petryashov, V.V., 2001. Chronology of the Holocene transgression at the North Siberian margin. *Global and Planetary Change* 31, 125-139.

Bauch, H.A., Polyakova, Y.I., 2003. Diatom-inferred salinity records from the Arctic Siberian Margin: Implications for fluvial runoff patterns during the Holocene. *Paleoceanography* 18, 1027.

Berben, S.M.P., Husum, K., Cabedo-Sanz, P., Belt, S.T., 2014. Holocene sub-centennial evolution of Atlantic water inflow and sea ice distribution in the western Barents Sea. *Clim Past* 10, 181-198.

Berger, G.W., 2006. Trans-arctic-ocean tests of fine-silt luminescence sediment dating provide a basis for an additional geochronometer for this region. *Quaternary Science Reviews* 25, 2529-2551.

Berger, G.W., 2009. Zeroing tests of luminescence sediment dating in the Arctic Ocean: Review and new results from Alaska-margin core tops and central-ocean dirty sea ice. *Global and Planetary Change* 68, 48-57.

Berger, G.W., 2011. Surmounting luminescence age overestimation in Alaska-margin Arctic Ocean sediments by use of ‘micro-hole’ quartz dating. *Quaternary Science Reviews* 30, 1750-1769.

Berger, G.W., Polyak, L., 2012. Testing the use of quartz ‘micro-hole’ photon-simulated luminescence for dating sediments from the central Lomonosov Ridge, Arctic Ocean. *Quaternary Geochronology* 11, 42-51.

Bintanja, R., van der Wiel, K., van der Linden, E.C., Reusen, J., Bogerd, L., Krikken, F., Selten, F.M., 2020. Strong future increases in Arctic precipitation variability linked to poleward moisture transport. *Science Advances* 6, eaax6869.

Broder, L., Tesi, T., Andersson, A., Semiletov, I., Gustafsson, O., 2018. Bounding cross-shelf transport time and degradation in Siberian-Arctic land-ocean carbon transfer. *Nature Communications* 9, 806.

- Chen, F., Chen, J., Huang, W., Chen, S., Huang, X., Jin, L., Jia, J., Zhang, X., An, C., Zhang, J., Zhao, Y., Yu, Z., Zhang, R., Liu, J., Zhou, A., Feng, S., 2019. Westerlies Asia and monsoonal Asia: Spatiotemporal differences in climate change and possible mechanisms on decadal to sub-orbital timescales. *Earth-Science Reviews* 192, 337-354.
- Comiso, J.C., Parkinson, C.L., Gersten, R., Stock, L., 2008. Accelerated decline in the Arctic sea ice cover. *Geophysical Research Letters* 35, L01703.
- Costard, F., Gautier, E., Fedorov, A., Konstantinov, P., Dupeyrat, L., 2014. An Assessment of the Erosion Potential of the Fluvial Thermal Process during Ice Breakups of the Lena River (Siberia). *Permafrost and Periglacial Processes* 25, 162-171.
- Darby, D.A., Bischof, J.F., Spielhagen, R.F., Marshall, S.A., Herman, S.W., 2002. Arctic ice export events and their potential impact on global climate during the late Pleistocene. *Paleoceanography* 17, 15-11-15-17.
- Darby, D.A., Myers, W.B., Jakobsson, M., Rigor, I., 2011. Modern dirty sea ice characteristics and sources: The role of anchor ice. *Journal of Geophysical Research* 116, C09008.
- Darby, D.A., Ortiz, J., Polyak, L., Lund, S., Jakobsson, M., Woodgate, R.A., 2009. The role of currents and sea ice in both slowly deposited central Arctic and rapidly deposited Chukchi–Alaskan margin sediments. *Global and Planetary Change* 68, 58-72.
- Darby, D.A., Ortiz, J.D., Grosch, C.E., Lund, S.P., 2012. 1,500-year cycle in the Arctic Oscillation identified in Holocene Arctic sea-ice drift. *Nature Geoscience* 5, 897-900.
- de Vernal, A., Hillaire-Marcel, C., Darby, D.A., 2005. Variability of sea ice cover in the Chukchi Sea (western Arctic Ocean) during the Holocene. *Paleoceanography* 20, PA4018.
- de Vernal, A., Hillaire-Marcel, C., Le Duc, C., Roberge, P., Brice, C., Matthiessen, J., Spielhagen, R.F., Stein, R., 2020. Natural variability of the Arctic Ocean sea ice during the present interglacial. *Proceedings of the National Academy of Sciences of the United States of America* 117, 26069-26075.
- de Vernal, A., Hillaire-Marcel, C., Rochon, A., Fréchette, B., Henry, M., Solignac, S., Bonnet, S., 2013. Dinocyst-based reconstructions of sea ice cover concentration during the Holocene in the Arctic Ocean, the northern North Atlantic Ocean and its adjacent seas. *Quaternary Science Reviews* 79,

111-121.

- Giraudeau, J., Grelaud, M., Solignac, S., Andrews, J.T., Moros, M., Jansen, E., 2010. Millennial-scale variability in Atlantic water advection to the Nordic Seas derived from Holocene coccolith concentration records. *Quaternary Science Reviews* 29, 1276-1287.
- Harada, N., Katsuki, K., Nakagawa, M., Matsumoto, A., Seki, O., Addison, J.A., Finney, B.P., Sato, M., 2014. Holocene sea surface temperature and sea ice extent in the Okhotsk and Bering Seas. *Progress in Oceanography* 126, 242-253.
- Hörner, T., Stein, R., Fahl, K., Birgel, D., 2016. Post-glacial variability of sea ice cover, river run-off and biological production in the western Laptev Sea (Arctic Ocean) – A high-resolution biomarker study. *Quaternary Science Reviews* 143, 133-149.
- Jakobsson, M., Backman, J., Murray, A., Løvlie, R., 2003. Optically Stimulated Luminescence dating supports central Arctic Ocean cm-scale sedimentation rates. *Geochemistry, Geophysics, Geosystems* 4, 1016.
- Karlsson, E., Gelting, J., Tesi, T., van Dongen, B., Andersson, A., Semiletov, I., Charkin, A., Dudarev, O., Gustafsson, Ö., 2016. Different sources and degradation state of dissolved, particulate, and sedimentary organic matter along the Eurasian Arctic coastal margin. *Global Biogeochemical Cycles* 30, 898-919.
- Klemm, J., Herzschuh, U., Pestryakova, L.A., 2016. Vegetation, climate and lake changes over the last 7000 years at the boreal treeline in north-central Siberia. *Quaternary Science Reviews* 147, 422-434.
- Klemm, J., Herzschuh, U., Pisaric, M.F.J., Telford, R.J., Heim, B., Pestryakova, L.A., 2013. A pollen-climate transfer function from the tundra and taiga vegetation in Arctic Siberia and its applicability to a Holocene record. *Palaeogeography, Palaeoclimatology, Palaeoecology* 386, 702-713.
- Kruppen, T., Janout, M., Hodges, K.I., Gerdes, R., Girard-Arduin, F., Hölemann, J.A., Willmes, S., 2013. Variability and trends in Laptev Sea ice outflow between 1992–2011. *The Cryosphere* 7, 349-363.
- Lammers, R.B., Pundsack, J.W., Shiklomanov, A.I., 2007. Variability in river temperature, discharge,

and energy flux from the Russian pan-Arctic landmass. *Journal of Geophysical Research: Biogeosciences* 112, G04S59.

Lindsay, R., Schweiger, A., 2015. Arctic sea ice thickness loss determined using subsurface, aircraft, and satellite observations. *The Cryosphere* 9, 269-283.

Locarnini, R.A., Mishonov, A.V., Antonov, J.I., Boyer, T.P., Garcia, H.E., Baranova, O.K., Zweng, M.M., Paver, C.R., Reagan, J.R., Johnson, D.R., Hamilton, M., Seidov, D., 2013. *World Ocean Atlas 2013, Volume 1: Temperature*. S. Levitus, Ed.; A. Mishonov, Technical Ed.; NOAA Atlas NESDIS 73, 40.

Magritsky, D., Alexeevsky, N., Aybulatov, D., Fofonova, V., Gorelkin, A., 2017. Features and evaluations of spatial and temporal changes of water runoff, sediment yield and heat flux in the Lena River delta. *Polarforschung* 87, 89-109.

Mann, P.J., Eglinton, T.I., McIntyre, C.P., Zimov, N., Davydova, A., Vonk, J.E., Holmes, R.M., Spencer, R.G., 2015. Utilization of ancient permafrost carbon in headwaters of Arctic fluvial networks. *Nature Communications* 6, 7856.

Marcott, S.A., Shakun, J.D., Clark, P.U., Mix, A.C., 2013. A reconstruction of regional and global temperature for the past 11,300 years. *Science* 339, 1198-1201.

Martens, J., Wild, B., Muschitiello, F., O'Regan, M., Jakobsson, M., Semiletov, I., Dudarev, O.V., Gustafsson, O., 2020. Remobilization of dormant carbon from Siberian-Arctic permafrost during three past warming events. *Science Advances* 6, eabb6546.

McCave, I.N., Andrews, J.T., 2019. Distinguishing current effects in sediments delivered to the ocean by ice. I. Principles, methods and examples. *Quaternary Science Reviews* 212, 92-107.

Morgenstern, A., Ulrich, M., Günther, F., Roessler, S., Fedorova, I.V., Rudaya, N.A., Wetterich, S., Boike, J., Schirrmeister, L., 2013. Evolution of thermokarst in East Siberian ice-rich permafrost: A case study. *Geomorphology* 201, 363-379.

Nürnberg, D., Wollenburg, I., Dethleff, D., Eicken, H., Kassens, H., Letzig, T., Reimnitz, E., Thiede, J., 1994. Sediments in Arctic sea ice: Implications for entrainment, transport and release. *Marine Geology* 119, 185-214.

Park, H., Watanabe, E., Kim, Y., Polyakov, I., Oshima, K., Zhang, X., Kimball, J.S., Yang, D., 2020.

Increasing riverine heat influx triggers Arctic sea ice decline and oceanic and atmospheric warming. *Science Advances* 6, eabc4699.

Perovich, D.K., Nghiem, S.V., Markus, T., Schweiger, A., 2007. Seasonal evolution and interannual variability of the local solar energy absorbed by the Arctic sea ice–ocean system. *Journal of Geophysical Research* 112, C03005.

Polyakov, I.V., Pnyushkov, A.V., Alkire, M.B., Ashik, I.M., Baumann, T.M., Carmack, E.C., Goszczko, I., Guthrie, J., Ivanov, V.V., Kanzow, T., Krishfield, R., Kwok, R., Sundfjord, A., Morison, J., Rember, R., Yulin, A., 2017. Greater role for Atlantic inflows on sea-ice loss in the Eurasian Basin of the Arctic Ocean. *Science* 356, 285-291.

Prange, M., Lohmann, G., 2003. Effects of mid-Holocene river runoff on the Arctic ocean/sea-ice system: a numerical model study. *The Holocene* 13, 335-342.

Rachold, V., 1999. Major, trace and rare earth element geochemistry of suspended particulate material of East Siberian rivers draining to the Arctic Ocean. *Land-Ocean Systems in the Siberian Arctic* Springer, Berlin, Heidelberg., 199-222.

Reimnitz, E., Clayton, J.R., Kempema, E.W., Payne, J.R., Weber, W.S., 1993. Interaction of rising frazil with suspended panicles: tank experiments with applications to nature. *Cold Regions Science and Technology* 21, 117-135.

Ruan, J., Huang, Y., Shi, X., Liu, Y., Xiao, W., Xu, Y., 2017. Holocene variability in sea surface temperature and sea ice extent in the northern Bering Sea: A multiple biomarker study. *Organic Geochemistry* 113, 1-9.

Stein, R., Fahl, K., Schade, I., Manerung, A., Wassmuth, S., Niessen, F., Nam, S.I., 2017. Holocene variability in sea ice cover, primary production, and Pacific-Water inflow and climate change in the Chukchi and East Siberian Seas (Arctic Ocean). *Journal of Quaternary Science* 32, 362-379.

Stranne, C., Jakobsson, M., Björk, G., 2014. Arctic Ocean perennial sea ice breakdown during the Early Holocene Insolation Maximum. *Quaternary Science Reviews* 92, 123-132.

Tananaev, N.I., 2016. Hydrological and sedimentary controls over fluvial thermal erosion, the Lena River, central Yakutia. *Geomorphology* 253, 524-533.

Taylor, S.R., McLennan, S.M., 1995. The geochemical evolution of the continental crust. *Reviews of*

Geophysics 33, 241-265.

- Tesi, T., Muschitiello, F., Smittenberg, R.H., Jakobsson, M., Vonk, J.E., Hill, P., Andersson, A., Kirchner, N., Noormets, R., Dudarev, O., Semiletov, I., Gustafsson, O., 2016. Massive remobilization of permafrost carbon during post-glacial warming. *Nature Communications* 7, 13653.
- Wagner, A., Lohmann, G., Prange, M., 2011. Arctic river discharge trends since 7 ka BP. *Global and Planetary Change* 79, 48-60.
- Wegner, C., Bennett, K.E., de Vernal, A., Forwick, M., Fritz, M., Heikkilä, M., Łacka, M., Lantuit, H., Laska, M., Moskalik, M., O'Regan, M., Pawłowska, J., Promińska, A., Rachold, V., Vonk, J.E., Werner, K., 2015. Variability in transport of terrigenous material on the shelves and the deep Arctic Ocean during the Holocene. *Polar Res* 34, 24964.
- West, G., Alexanderson, H., Jakobsson, M., O'Regan, M., 2021. Optically stimulated luminescence dating supports pre-Eemian age for glacial ice on the Lomonosov Ridge off the East Siberian continental shelf. *Quaternary Science Reviews* 267, 107082.
- Winterfeld, M., Mollenhauer, G., Dumann, W., Kohler, P., Lembke-Jene, L., Meyer, V.D., Hefter, J., McIntyre, C., Wacker, L., Kokfelt, U., Tiedemann, R., 2018. Deglacial mobilization of pre-aged terrestrial carbon from degrading permafrost. *Nature Communications* 9, 3666.
- Yang, D., Kane, D.L., Hinzman, L.D., Zhang, X., Zhang, T., Ye, H., 2002. Siberian Lena River hydrologic regime and recent change. *Journal of Geophysical Research: Atmospheres* 107, 4694.
- Yang, D., Shrestha, R.R., Lung, J.L.Y., Tank, S., Park, H., 2021. Heat flux, water temperature and discharge from 15 northern Canadian rivers draining to Arctic Ocean and Hudson Bay. *Global and Planetary Change* 204, 103577.
- Zhang, X., He, J., Zhang, J., Polyakov, I., Gerdes, R., Inoue, J., Wu, P., 2012. Enhanced poleward moisture transport and amplified northern high-latitude wetting trend. *Nature Climate Change* 3, 47-51.
- Zweng, M.M., Reagan, J.R., Antonov, J.I., Locarnini, R.A., Mishonov, A.V., Boyer, T.P., Garcia, H.E., Baranova, O.K., Johnson, D.R., Seidov, D., Biddle, M.M., 2013. *World Ocean Atlas 2013, Volume 2: Salinity*. Levitus, Ed.; A. Mishonov, Technical Ed.; NOAA Atlas NESDIS 74, 39.

Reviewers' Comments:

Reviewer #1:

Remarks to the Author:

I commend you on addressing the reviewer's concerns.

Reviewer #2:

Remarks to the Author:

I would like to commend the authors on doing a very thorough job in revising this manuscript. They have adequately considered and addressed all of my comments in the revision.

I believe they have compiled a large amount of existing multiproxy data, and integrated this with their new data to successfully highlight how an important feedback between the warming and discharge of freshwater from Arctic rivers and sea ice conditions in a 'warmer' Arctic. The manuscript provides a significant advance on our current state of knowledge, that will certainly motivate considerable future research.

I have a few somewhat critical comments on the wording of certain phrases in the manuscript. These are outlined in the annotated PDF. I think these need to be fixed prior to publication. More generally, the manuscript needs a thorough and final proof-reading by the authors to correct grammar and some confusing sentences.

Two small things I think should very briefly be highlighted are 1) that increased rates of sedimentation in the Late Holocene have been documented before for the Laptev and ESS shelves and slopes. However, this HAS NOT been interpreted as a response to enhanced river discharge - but moreover as a response to sealevel rise. That should be made clear. Sedimentation rates are not an established proxy for river derived sediment delivery to Arctic shelves - that is a unique argument of this paper. 2) Following on from this, a final supplementary figure the authors could/should include, is how the 'shoreline' migrated during sea-level rise in the Middle to Late Holocene. They argue that global sea level rise was minor - but the region of the ESS shallower than 20-30 m is enormous. If they can show that the core positions were not dramamtically influenced by this, or alternatively, show how they would have been impacted and argue why this would not influence the grain size records, I believe it would add a final bit of rigour to their arguments. I do not think it can explain the increase in coarse fraction content that they see in their records - so it should support their arguments for more sea ice). For example - intuitively, one would think that as the core sites moved into deeper water depths, the coarse fraction content of the sediments would decrease (as is well documented on the Beaufort shelf (Jerosch, K. 2013. Geostatistical mapping and spatial variability of surficial sediment types on the Beaufort Shelf based on grain size data. J. Mar. Syst. 2013, 127, 5-13). This could be added to the discussion with a nice supporting figure showing very basic snapshots of the predicted coast at snapshots in the MH and LH.

In conclusion, my opinion would be to accept the manuscript following some very minor revisions.

Sincerely,
Matt O'Regan

Response to Reviewer #1:

I commend you on addressing the reviewer's concerns.

Thanks for all the help on the manuscript!

Response to Reviewer #2:

I would like to commend the authors on doing a very thorough job in revising this manuscript. They have adequately considered and addressed all of my comments in the revision.

I believe they have compiled a large amount of existing multiproxy data, and integrated this with their new data to successfully highlight how an important feedback between the warming and discharge of freshwater from Arctic rivers and sea ice conditions in a 'warmer' Arctic. The manuscript provides a significant advance on our current state of knowledge, that will certainly motivate considerable future research.

I have a few somewhat critical comments on the wording of certain phrases in the manuscript. These are outlined in the annotated PDF. I think these need to be fixed prior to publication. More generally, the manuscript needs a thorough and final proof-reading by the authors to correct grammar and some confusing sentences.

Thanks for your comments, in both scientific and linguistic aspects. In this-around revision, we've carefully done a linguistic check, based on the annotated PDF and also the help of native English-speaker specialists.

Two small things I think should very briefly be highlighted are

1) that increased rates of sedimentation in the Late Holocene have been documented before for the Laptev and ESS shelves and slopes. However, this HAS NOT been interpreted as a response to enhanced river discharge - but moreover as a response to sealevel rise. That should be made clear. Sedimentation rates are not an established proxy for river derived sediment delivery to Arctic shelves - that is a unique argument of this paper.

We agree with the reviewer that using sediment rate as a proxy for the river discharge in this work is new to the existing understanding of the changing distance to the terrestrial sediment sources along sea level rise over the East Siberian Arctic Shelf during the Holocene (e.g. Wegner et al. 2015). Mechanically, river discharge and sea-level induced distance between the coast and core location are two regulators working together in changing the sediment rate in the river-impact marginal seas, i.e. the East Siberian Arctic Shelf in the Holocene. Here, following this comment and the note for lines 74-75 in the annotated PDF, we've adjusted this sentence to 'The sedimentation rate in the East Siberian Arctic Shelf region during the mid-to-late Holocene acts as a potential proxy of the river

discharge, adding to the impact of sea level change'. Please see lines 74-78 in the revised Main Text.

2) *Following on from this, a final supplementary figure the authors could/should include, is how the 'shoreline' migrated during sea-level rise in the Middle to Late Holocene. They argue that global sea level rise was minor - but the region of the ESS shallower than 20-30 m is enormous. If they can show that the core positions were not dramatically influenced by this, or alternatively, show how they would have been impacted and argue why this would not influence the grain size records, I believe it would add a final bit of rigour to their arguments. I do not think it can explain the increase in coarse fraction content that they see in their records - so it should support their arguments for more sea ice). For example - intuitively, one would think that as the core sites moved into deeper water depths, the coarse fraction content of the sediments would decrease (as is well documented on the Beaufort shelf (Jerosch, K. 2013. Geostatistical mapping and spatial variability of surficial sediment types on the Beaufort Shelf based on grain size data. J. Mar. Syst. 2013, 127, 5–13). This could be added to the discussion with a nice supporting figure showing very basic snapshots of the predicted coast at snapshots in the MH and LH.*

This suggestion is similar to the reviewer's comment on Line 89-90 in the annotated pdf. Here, we agree with the reviewer and have newly added Supplementary Figure 10 for the sea level change in the East Siberian Sea. According to paleoceanographic reconstructions, the sea level has reached the present-day state since ~7 ka. During the mid-Holocene (MH, 7.5–4.0 ka), the sea level was -0.75 ± 4.55 m, with few reconstruction points showing outliers up to ~ -10 m, while it was 0.25 ± 2.45 m in the late Holocene (4.0–0 ka) relative to the present. As shown in Supplementary Fig. 10, the paleo-coastal line has been relatively steady since the MH. This thus echos a case of the reviewer's thought that our core positions were not dramatically influenced by sea level since the MH. We've added the citation of new Supplementary Fig. 10 in lines 92-96 of the revised manuscript.

Newly added Supplementary Fig. 10 Paleo-sea levels in the East Siberian Arctic Shelf (ESAS) region, according to Lambeck et al., 2014. The blue and yellow areas show the potential positions of paleo-coastlines during the mid-Holocene (MH) and late Holocene (LH), respectively. The red line is the location of the present-day coastline. The dashed blue line point to the outlier of the reconstructed sea level during the MH.

Comments in the annotated PDF, lines 74-75: The way this sentence is written makes it sound like sedimentation rates are an established proxy for river inflow or its thermal state. This is not correct. Simply look at the compilation of sedimentation rates in Wegner et al 2015 - and their explanation (a response to sea level rise). The authors recognise this in the response letter, and should do so in the manuscript. They provide a different interpretation of why sedimentation rates increase, not use an established one.

Of course this is fine, and I think they do a great job at justifying this in their work. It is a unique argument of this paper, and not an established 'proxy'.

'The regional sedimentation rate is a potential proxy of river thermal flux'

We agree with the reviewer, and have rephrased this part to 'The sedimentation rate in the ESAS region since the MH, which is primarily controlled by the river material supply, acts as a potential

proxy of the river thermal discharge, adding to the impact of sea level change'. Please see lines 76-78 in the revised Main Text.

Comments in the annotated PDF, Line 134 regnant export drainage → I am not sure what this means

Here, 'regnant export drainage' means the major transport route of thawed permafrost from inland to marginal seas, and we've changed it accordingly. Please see lines 141-144.

In addition, all your comments on the grammar and wording, as listed in the annotated PDF, have been accepted by us and adapted into the revised manuscript. Thanks a lot!

In conclusion, my opinion would be to accept the manuscript following some very minor revisions.

Thanks again for all the help!